# Capturing change in clonal composition amongst single mouse germinal centers

**Daniel J Firl[1,2,3], Soren E Degn[2,4], Timothy Padera[5], Michael C Carroll[2,6]***

[1]Cleveland Clinic Lerner College of Medicine, Cleveland, United States; [2]Program in Cellular and Molecular Medicine, Boston Children's Hospital, Boston, United States; [3]Howard Hughes Medical Institute, Maryland, United States; [4]Department of Biomedicine, Aarhus University, Aarhus, Denmark; [5]Edwin L. Steele Laboratories for Tumor Biology, Department of Radiation Oncology, Massachusetts General Hospital, Boston, United States; [6]Department of Pediatrics, Harvard Medical School, Boston, United States

**Abstract** Understanding cellular processes occurring in vivo on time scales of days to weeks requires repeatedly interrogating the same tissue without perturbing homeostasis. We describe a novel setup for longitudinal intravital imaging of murine peripheral lymph nodes (LNs). The formation and evolution of single germinal centers (GCs) was visualized over days to weeks. Naïve B cells encounter antigen and form primary foci, which subsequently seed GCs. These experience widely varying rates of homogenizing selection, even within closely confined spatial proximity. The fluidity of GCs is greater than previously observed with large shifts in clonality over short time scales; and loss of GCs is a rare, observable event. The observation of contemporaneous, congruent shifts in clonal composition between GCs within the same animal suggests inter-GC trafficking of memory B cells. This tool refines approaches to resolving immune dynamics in peripheral LNs with high temporospatial resolution and minimal perturbation of homeostasis.
DOI: https://doi.org/10.7554/eLife.33051.001

**\*For correspondence:**
michael.carroll@childrens.harvard.edu

**Competing interests:** The authors declare that no competing interests exist.

## Introduction

The advent of intravital imaging techniques allowing sequential, continuous imaging of tissues in the live organism for up to 8–12 hr has significantly advanced several fields over the past decade, not least within immunology (*Allen et al., 2007*; *Miller et al., 2002*; *Roozendaal et al., 2009*; *Schwickert et al., 2007*). Initially confined to peripherally accessible tissues, the scope of targetable locations has recently expanded notably to include for example the spleen in the peritoneal cavity (*Arnon et al., 2013*), and even the heart within the thoracic cavity (*Vinegoni et al., 2015*). However, a significant gap has remained between the capability for continuous imaging of individual animals on time scales of less than half a day, and the statistically based analysis of timed cohorts of experimental animals on longer time scales. This gap has narrowed in the field of neurobiology by the advent of techniques such as the cranial (*Holtmaat et al., 2009*) and the thinned skull (*Yang et al., 2010*) windows, and similar methodologies have been harnessed for long-term imaging of tumors, such as the mammary window (*Kedrin et al., 2008*), and the abdominal imaging window (*Ritsma et al., 2013*). Transecting the fields of tumor biology and immunology, a discontinuous longitudinal imaging model was recently applied to study dissemination of lymphoma in the murine lymph node (LN) (*Ito et al., 2012*), and the lack of angiogenesis in the growth of LN metastasis was investigated using a modification of the chronic mammary fat pad window model (*Jeong et al., 2015*). A model of LN transplantation to the murine ear and subsequent imaging has also been reported (*Gibson et al., 2012*). Notwithstanding these efforts, a full-fledged setup for longitudinal imaging of complex cellular dynamics in LNs in living animals has been lacking. Here we present

such a setup, and its application to the central immunological question of initiation and clonal evolution of single germinal centers (GCs) in response to foreign, as well as self-antigen.

Adaptive immune responses are crucial to long-lived immunological memory, providing more rapid and effective clearance of previously encountered pathogens (*Kurosaki et al., 2015*; *McHeyzer-Williams et al., 2011*). Following antigen drainage and transport to the LN, cognate naïve follicular B cells recognize antigen and become activated through B cell receptor (BCR) cross-linking and signaling (*Barnett et al., 2014*; *De Silva and Klein, 2015*). Concomitantly, cognate T cells recognize antigen-derived peptides presented by DCs and they take on the characteristics of T follicular helper cells (Tfh), allowing them to provide a necessary second signal to begin an adaptive immune response (*Barnett et al., 2014*). Following antigen engagement and T cell help, cognate B cells seed early GCs (*De Silva and Klein, 2015*; *Heesters et al., 2014*). In the GCs they interact with follicular dendritic cells (FDCs), stromal cells that act as long-lived reservoirs of antigen (*Heesters et al., 2013*) and may contribute cytokines to shape the response (*Das et al., 2017*), and with Tfh (*Shulman et al., 2014*). B cell clones which successfully take up and present antigen to Tfh receive survival and proliferation signals. Establishment of the GC response involves expansion of dozens to hundreds of cognate B cell centroblasts, which cycle between light and dark zones (*Tas et al., 2016*; *Victora et al., 2010*). In an iterative process of division and diversification through somatic hypermutation (SHM) in the dark zone (DZ), and affinity-dependent antigen probing and presentation in the light zone (LZ), Darwinian selection drives the process of affinity maturation (*De Silva and Klein, 2015*; *Heesters et al., 2014*; *Hauser et al., 2010*; *Beltman et al., 2011*).

During commitment to the GC phenotype, B cells express activation-induced cytidine deaminase (*Aicda*), which is the central enzyme responsible for the process of SHM (*Allen et al., 2007*; *De Silva and Klein, 2015*; *Heesters et al., 2014*; *Victora et al., 2010*; *Beltman et al., 2011*). In recent years, numerous genetic multicolor systems have been developed for lineage tracing studies, broadly referred to as 'Brainbow' or 'Confetti' systems (*Weissman and Pan, 2015*; *Bonaguidi et al., 2011*; *Centanin et al., 2014*). Recently, an Aicda-CreERT2 driver line was combined with the Confetti system, allowing visualization of GC selection in vivo (*Tas et al., 2016*). Combining time point analyses with population-modeling, it was found that GC responses are started by as little as a few hundred naïve B cells reacting to the challenging antigen and that GCs proceed towards pauciclonality at differential rates. Here, we harness the power of this experimental system by using longitudinal intravital imaging and further explore the mechanisms behind GC formation and shifts in clonality.

Hypothesizing that GC dynamics can be better resolved by following single GC over time, increasing the power of observations exponentially, we combined microsurgical implantation of an imaging window onto the inguinal LN (iLN) with gentle multiphoton imaging and the power of the Confetti reporter. The early events of GC initiation were followed by tracking B cells migrating to and between early GC and primary foci. We also measured the dynamics of B cell clonality at the single GC level over time.

## Results

### The Chronic lymph node window (CLNW) model

To allow longitudinal studies of cellular dynamics in the LN, we adopted a modification of the chronic mammary fat pad window model referred to as the chronic lymph node window (*Jeong et al., 2015*). Using sterile surgical technique, window chambers were surgically implanted over the iLN, and a small incision made in the skin to expose the underlying node (*Figure 1*). The chambers were sealed with a coverslip and O-ring (*Figure 1D*) to prevent contamination of the field by external contaminants, further mice received one week of perioperative antibiotics. Following implantation, mice were periodically anaesthetized and imaged using multi-photon microscopy with a customized setup (Materials and methods and *Figure 1E*).

### Homeostatic conditions are maintained in the CLNW model

To ensure that the surgery and window implantation did not significantly affect any observations made, several control experiments were performed. The percent change in body weight was followed in a group of CLNW implanted mice and a control group that was sham operated (skin opened, tissue over the node removed and then skin closed, recovered as standard) but did not

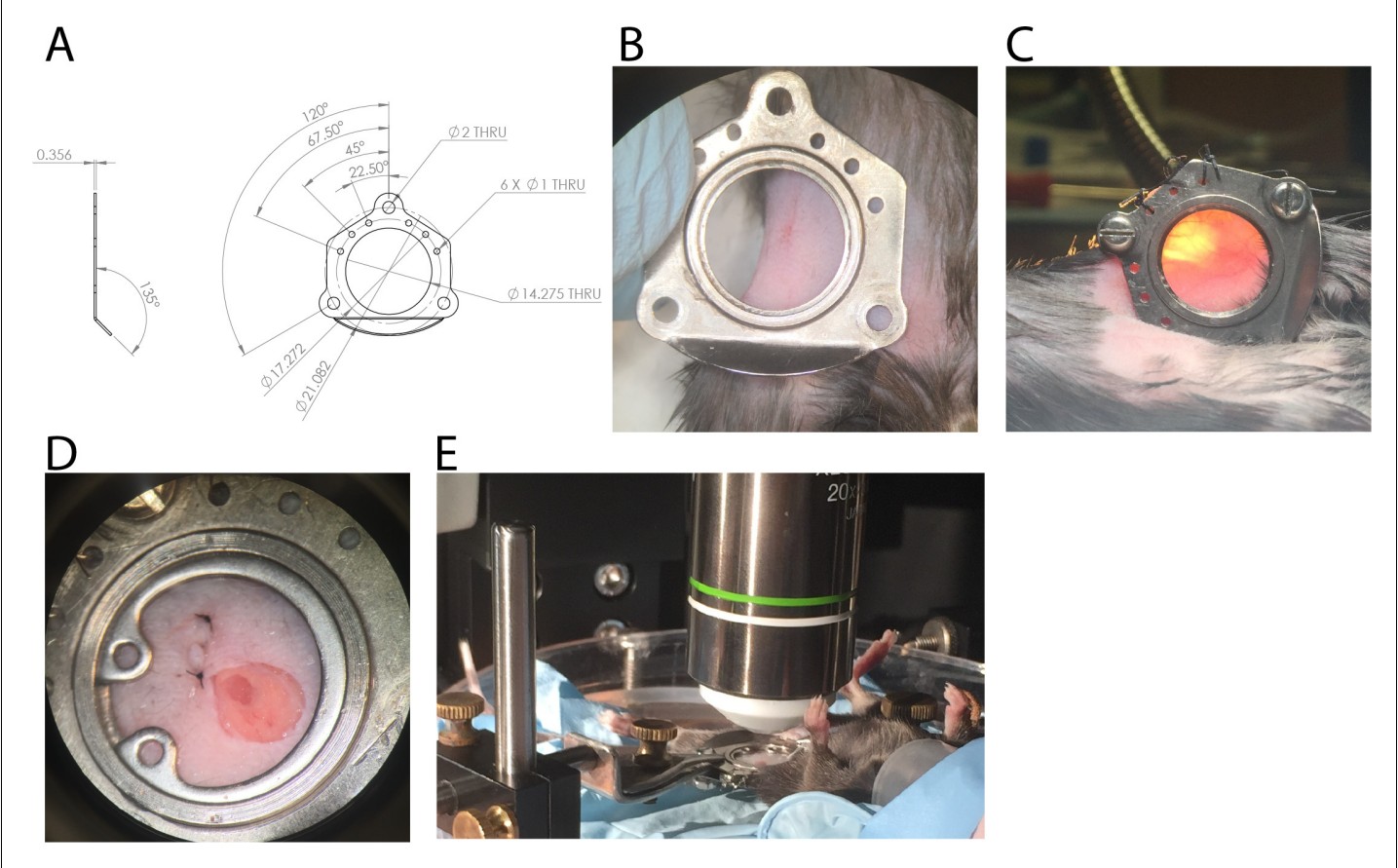

**Figure 1.** Design and surgical implantation of titanium CLNW. (**A**) CAD schematic detailing the dimensions of the titanium plates opposed to fit a chronic lymph node window (CLNW). (**B**) Following shaving the surgical site, window plates are briefly overlaid for surgical planning and prior to sterile preparation. (**C**) Following surgical implantation of the CLNW, transillumination confirms the node centered appropriately within the imaging range. (**D**) Photomicrograph of completed CLNW demonstrating coverglass, incision closure, and LN placement within the center of viewing range. (**E**) Imaging setup demonstrating anesthetized mouse on a heated $H_2O$ reservoir with clamp fixation of the window minimizing motion artifact.

DOI: https://doi.org/10.7554/eLife.33051.002

The following figure supplement is available for figure 1:

**Figure supplement 1.** Additional surgical images demonstrates anchor placement and node exposure.

DOI: https://doi.org/10.7554/eLife.33051.003

receive the window implantation. As seen in *Figure 2A*, CLNW bearing mice experienced a slightly greater initial weight loss following the surgical procedure than the controls. However, by Day 3, they had rebounded to preoperative weight, and subsequently followed the same weight course development as control animals (p=0.107 for treatment in two-way ANOVA) (*Figure 2A*). We also verified intact and sufficient lymphatic drainage to the implanted node by injecting and measuring PE-IC complexes, either before or after window implantation (*Figure 2—figure supplement 1*). Absence of fluid accumulation and swelling in the surrounding tissue confirmed adequate lymphoid drainage from the site. Following conclusion of the window implantation (14 days), mice were sacrificed, and the window implanted iLN was compared with the contralateral iLN. No gross histological differences (size, shape, vascularization) were noted (representative examples in *Figure 2B*). Absolute numbers of immune cell subsets were quantified by flow cytometric analyses of spleen (*Figure 2C*), mesenteric LN (mLN, *Figure 2D*), the contralateral iLN (*Figure 2E*), and the CLNW implanted iLN (*Figure 2F*), as well as the brachial LN (bLN) (*Figure 2G*). There was a decrease in relative T cell composition (CD4 and CD8) with increase in B cell (B220) composition. A trend towards an opposite effect was observed in the brachial LN which may indicate an effect on systemic lymphatic flow as opposed to true inflammatory processes. Additionally, there were slight increases in

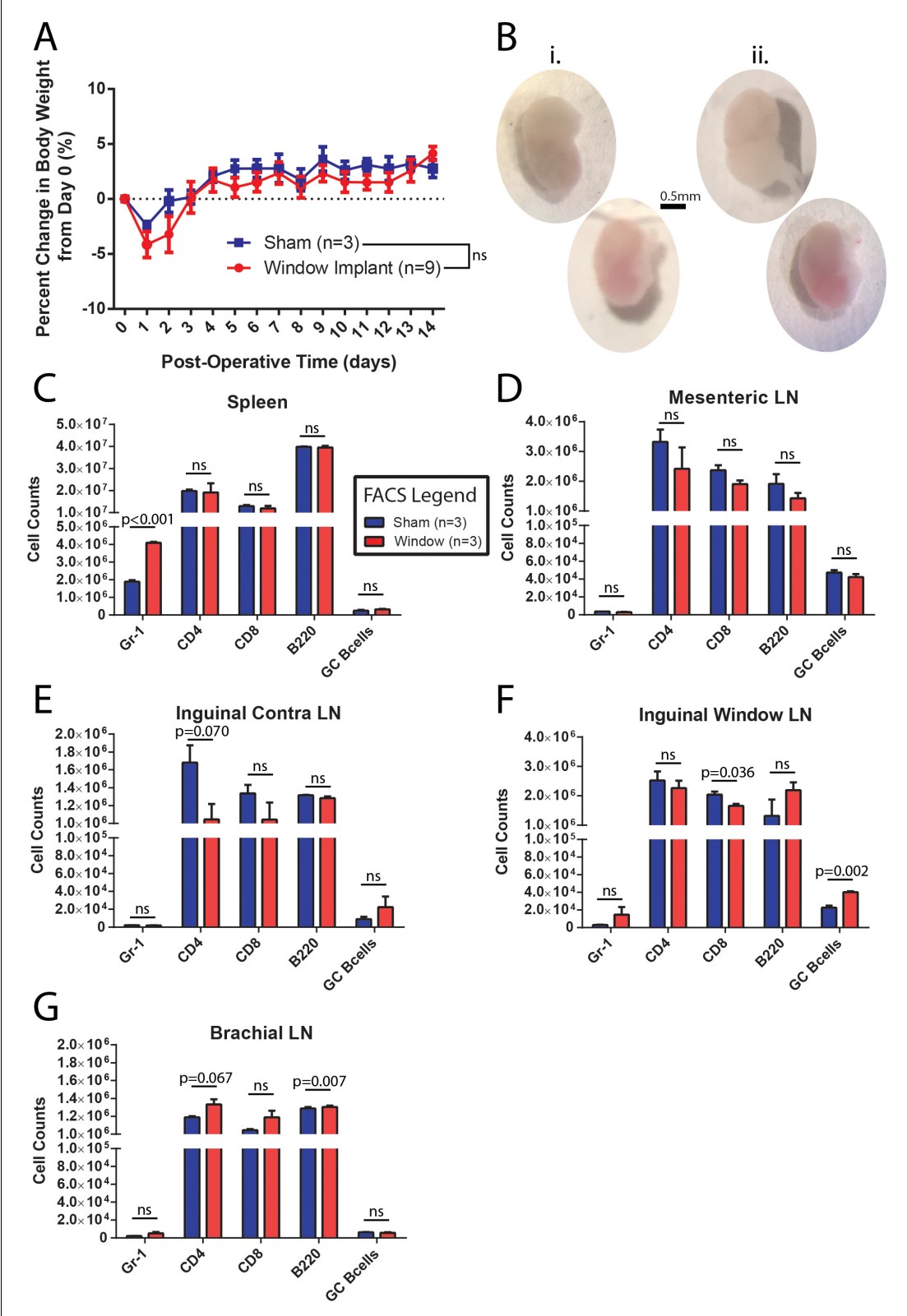

**Figure 2.** CLNW implantation does not dramatically perturb homeostasis. (**A**) Daily weight monitoring of sham operated and window implanted mice demonstrates a slightly more dramatic post-operative decline followed by correction to the level of controls by day 3. Experiments conducted on more than three independent occasions. (**B**) Gross morphological assessment following 2 weeks within the CLNW. The sham operated LNs [I; two vertical nodes at left] were found to be similar to imaged nodes [ii; two vertical nodes at right] on the basis of size, vascularity, and consistency. (**C**) Flow

*Figure 2 continued on next page*

*Figure 2 continued*

cytometric quantification of numbers of live, CD45+ gate for Gr-1+, CD4+, CD8+, B220+, and GL7$^{hi}$CD38$^{lo}$ GC B cells in spleen compared between sham operated and CLNW implanted mice (n = 3 per group). Mean ± SEM indicated. P values obtained from chi squared analysis of proportions of live CD45+ cells. (D) Similar to (C) except for mesenteric lymph nodes. (E) Similar to (C) except for the contralateral (opposite operated side) inguinal lymph nodes. (F) Similar to (C) except for CLNW implanted inguinal lymph nodes. (G) Similar to (C) except for brachial lymph nodes.

DOI: https://doi.org/10.7554/eLife.33051.004

The following source data and figure supplement are available for figure 2:

**Source data 1.** CLNW and control mice weights and cell counts.
DOI: https://doi.org/10.7554/eLife.33051.006
**Source data 2.** CLNW and control mean pixel intensity for perioperative PE-IC lymphatics assessment.
DOI: https://doi.org/10.7554/eLife.33051.007
**Figure supplement 1.** Perioperative PE-IC immunization reveals intact lymphatics in CLNW operated mice.
DOI: https://doi.org/10.7554/eLife.33051.005

Gr1 positive cells, representing both Ly6C (monocyte) and Ly6G (neutrophil) subsets in the spleen and window implanted iLN, and a small increase in GC B cells (*Figure 2C and F*). These changes could potentially be attributed to a low-level sterile inflammation associated with disruption of the overlying tissue and exposure of the node, whereas they appeared too mild to be caused by contamination of the window site and unlikely related to antibiotics as sham operated mice also received them. We concluded that CLNW implantation did not cause a dramatic perturbation of homeostatic conditions.

## Initiation of germinal centers

To investigate the initiation of GCs, naïve B cells were purified from spleen and lymph nodes of B1-8hi CFP donors and transferred into CCL19-Cre EYFP mice (displaying YFP labeling of fibroblastic reticular cells (FRCs) as a 'counterstain' [*Cremasco et al., 2014*]) at Day −2. Then recipients were immunized with 15 µg NP-CGG subcutaneously in the groin and footpad bilaterally on Day 0, and window implantation was performed on Day 2. Mice were imaged every 24 hr starting from 48 hr post immunization, and the frequency of CFP+ cells per follicle was quantified (*Figure 3A and B*). Paracortical CFP+ cells were seen as early as 48–72 hr in very small numbers (*Figure 3C*) and small primary foci were observable around 72–96 hr (*Figure 3D*). Interestingly, on more than one occasion between 72–120 hr, CFP+ cells were seen to traffic between the core, high density GC and the paracortical clusters (*Figure 3—video 1*). This may demonstrate simple continuation of primary foci efflux or early trafficking between the primary foci and GCs. This movement ceased as the paracortical clusters dissipated 120–168 hr into the response. By 120 hr, GC were seeded, and within these, CFP + cells were seen to expand at an exponential rate over the subsequent days.

The B1-8hi CFP+GC initiation in response to NP-CGG was modelled by fitting exponential growth functions to the observed data (*Table 1*). The CFP+ cells in five out of eight GC conformed to the predicted response dynamics (*Figure 3—video 2*), displaying doubling times on the order of 6–12 hr, in line with what has been previously reported (Fo1, 2, 4, 6, 7) (*Anderson et al., 2009*). CFP + cells in the remaining three GC (Fo3, 5, 8) displayed much longer doubling times, reflective either of their failure to establish a GC altogether or to become GC winners. Notably these also expressed poor fit statistics. The average of the GCs observed conformed to the expected results based on earlier investigations, but the longitudinal approach allowed identification of a fraction (Fo3, 5, 8; 3/8, 37.5%) of non-conforming events, which would not have been observable with traditional time-point analyses. The observation of follicles in which CFP+ cells were present early on, expanded until around 132 hr, but then subsequently waned, was remarkable. As indicated, this could be caused by competition with a dark clone (from the endogenous repertoire) or, perhaps less likely, a failure to establish productive GC. The former possibility would suggest very early competition between B cell clones that occurs already before or during seeding of GCs.

## GC evolution is accurately reflected in the CLNW model

Although our data suggested that overall homeostasis was maintained in the CLNW model, the question remained whether longer term GC evolution could be perturbed. To investigate this, we employed a novel chimeric model of spontaneous autoreactive GC formation (the ARTEMIS model,

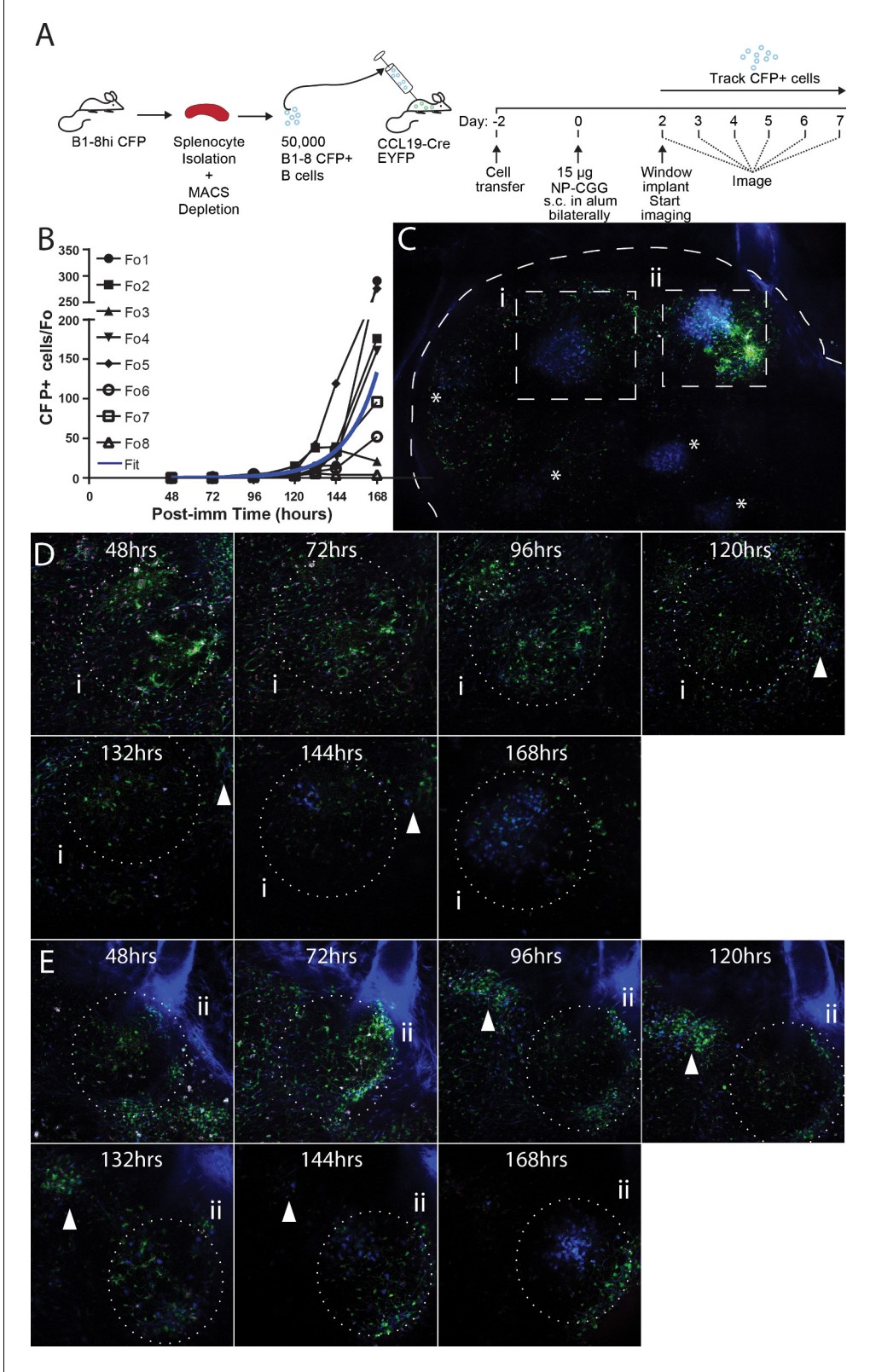

**Figure 3.** Observing the establishment of GC to foreign antigen using the CLNW. (**A**) Schematic outlining the adoptive transfer of B-18hi cells into CCL19-CreEYFP recipients on day −2, the immunization with NP-CGG bilaterally on day 0 and surgical implantation of the CLNW on day two with subsequent daily imaging. Experiments conducted on two independent preparations with two independent mice. (**B**) Graphical quantification of CFP + cells within longitudinally observed single follicles from days 2–7 demonstrating an average exponential increase in cellularity with a surprising level of

*Figure 3 continued on next page*

*Figure 3 continued*

variability. Individual black lines with different symbols for each Fo as in the legend, blue overlay is the average fit line. (C) Following 7 days of imaging, a single CLNW is observed at constant depth (~150 um) and images 'stitched' to form a GC map. *represent the location of GC, of varying intensity due to depth. Second harmonics generation from collagen rendered in blue, YFP+ cells rendered in green and CFP+ cells rendered in blue. (D) Follicle (i) from empty through evolution to a germinal center over the 48–168 hr observation, shown in (C), white arrowhead demonstrates location of extafollicular CFP+ cells. (E) Follicle (ii) from empty through evolution to a germinal center over the 48–168 hr observation, shown in (C), white arrowhead demonstrates location of extafollicular CFP+ cells.

DOI: https://doi.org/10.7554/eLife.33051.008

The following video and source data are available for figure 3:

**Source data 1.** Early GC formation CFP+ cell counts.
DOI: https://doi.org/10.7554/eLife.33051.009
**Figure 3—video 1.** Intravital imaging of CFP+ B1-8 hi cells in in NP-CGG immunized, CCL19-EYFP reporter.
DOI: https://doi.org/10.7554/eLife.33051.010
**Figure 3—video 2.** Intravital imaging of CFP+ B1-8 hi cells in in NP-CGG immunized, CCL19-EYFP reporter with PE-immune complex immunization for FDC visualization.
DOI: https://doi.org/10.7554/eLife.33051.011

[*Degn et al., 2017*]) or foreign-antigen NP-CGG immunization (imm). Of note, although these models are very different, one reflecting autoreactive responses to complex antigen, the other a response to a less complex foreign antigen, they were found to have overall similar clonal evolution kinetics by extensive conventional time-point analyses using the Confetti model (*Degn et al., 2017*). For autoreactive GC, following reconstitution, chimeric mice (n = 2) were tamoxifen treated to switch on the confetti reporter, and then received CLNW implants. For foreign antigen GC, Confetti mice (n = 2) were immunized subcutaneously with NP-CGG, were tamoxifen treated, and then received CLNW implants (*Figure 4A*). At specified time points, the window implanted iLNs were imaged, and then the mice were sacrificed and the contralateral iLN explanted and imaged. For each paired window iLN and contralateral explanted iLN, the average of the frequency of most dominant clone (*Figure 4B–C*), sum of most and second-most dominant clones (*Figure 4D*), and divergence index (*Figure 4E*), were calculated and correlation plots were generated. Example paired images can be seen in *Figure 4F–K* for window (top) and explant (bottom) LNs. Although depth resolution and image quality was inferior in vivo, compared to explant, color dominance was accurately quantifiable. As can be seen, based on all three parameters, windows and explants correlated well, with the slope approximating 1. For example, for the NP-CGG response, we found similar mean clonality at day 11 and day 25 post tamoxifen treatment regardless of whether GCs were analyzed via explant or by longitudinal imaging (Day 11 post imm 44.8% vs 42.8% and Day 25 post imm 77.7% vs 80.3%, respectively; p=0.890). Therefore, the CLNW was not found to adversely affect the GC dynamics and clonal evolution.

**Table 1.** Modeling* the B1-8hi CFP+ GC Initiation to NP-CGG.

| | $Y_0$ | K | Tau | Doubling time (hours) | $R^2$ |
|---|---|---|---|---|---|
| Fo1 | $2.5 \times 10^{-6}$ $(-8.5 \times 10^{-6} - 1.4 \times 10^{-5})$ | 0.110 (0.085–0.136) | 9.06 (7.35–11.81) | 6.28 (5.10–8.18) | 0.9980 |
| Fo2 | 0.022 (−0.022–0.066) | 0.053 (0.042–0.065) | 18.73 (15.32–24.09) | 12.98 (10.62–16.70) | 0.9890 |
| Fo3 | 1.469 (−3.945–6.883) | 0.018 (−0.006–0.0422) | 56.49 (23.68-inf) | 39.15 (16.41-INF) | 0.5811 |
| Fo4 | 0.001 (0.001–0.001) | 0.072 (0.069–0.753) | 13.82 (13.28–14.40) | 9.58 (9.21–9.99) | 0.9998 |
| Fo5 | 0.120 (−0.147–0.387) | 0.046 (0.033–0.060) | 21.66 (16.74–30.67) | 15.01 (11.61–21.26) | 0.9781 |
| Fo6 | 0.005 (−0.005–0.015) | 0.055 (0.043–0.060) | 18.13 (14.91–23.13) | 12.57 (10.34–16.03) | 0.9895 |
| Fo7 | 0.010 (−0.016–0.036) | 0.055 (0.039–0.071) | 18.26 (14.11–25.88) | 12.66 (9.78–17.94) | 0.9820 |
| Fo8 | 0.472 (−0.873–1.817) | 0.014 (−0.005–0.034) | 69.67 (29.58-inf) | 48.29 (20.50-INF) | 0.5779 |
| Average | 0.011 (0.006–0.016) | 0.056 (0.053–0.059) | 17.86 (17.08–18.70) | 12.38 (11.84–12.96) | 0.9996 |

DOI: https://doi.org/10.7554/eLife.33051.020

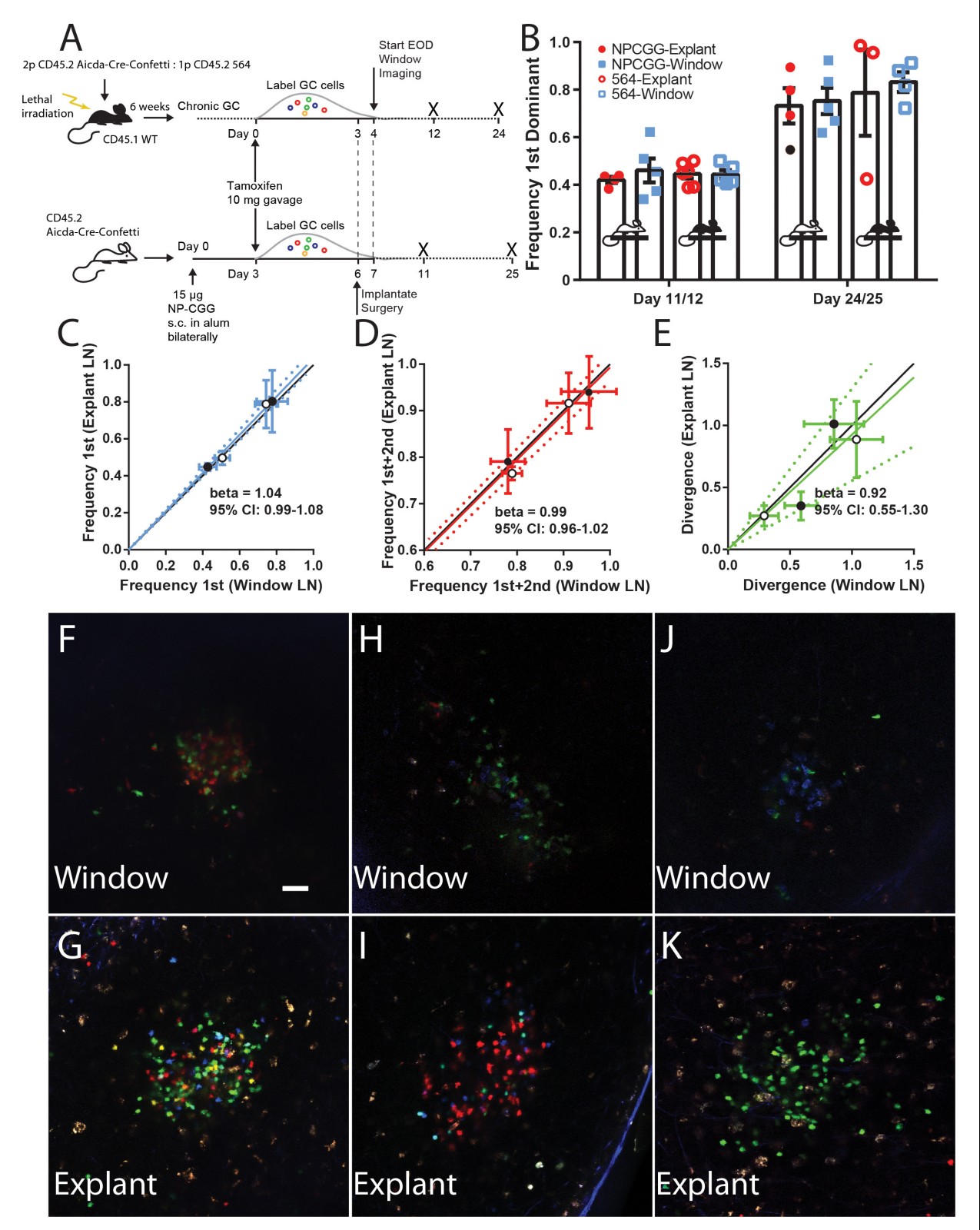

**Figure 4.** Establishing fidelity of clonal development in CLNW observed GC. (**A**) Schematic overview of experimental approach for generation and analysis of CD45.2 *Aicda*-Cre-Confetti and CD45.2 *Aicda*-Cre-Confetti 564 mixed BM chimeras. Experiments conducted on two independent preparations with two independent mice each. (**B**) Overview of fidelity of the window imaging model. The frequency of most (first) dominant colors observed in individual mice between GCs in the CLNW as compared to the explanted contralateral LN observed ex vivo. These observations are shown

*Figure 4 continued on next page*

*Figure 4 continued*

in both the average (bar with standard error) as well as individual values plotted per the key for both NPCGG and 564 at early (day 11/12) and later (day 24/25) times. (**C**) Comparing the frequency of most (first) predominant colors observed in individual mice between GCs in the CLNW as compared to the contralateral LN observed ex vivo. Both NP-CGG and 564 comparisons are included and a linear regression is fitted, demonstrating a beta coefficient overlapping with unity, 1.04 (0.99–1.08). The mean of an average of 4.1 GC per LN for each mouse at each time point, for both NP-CGG and 564, is indicated by each point. (**D**) Similar to (**C**) except comparing the frequency of the most and second most predominant colors as observed ex vivo from the contralateral LN and in vivo in the CLNW. Regression line with beta coefficient overlapping unity, 0.99 (0.96–1.02). (**E**) Similar to (**C**) except comparing the divergence index for color composition by GC as observed ex vivo from the contralateral and in vivo in the CLNW. Regression line with beta coefficient overlapping unity, 0.92 (0.55–1.30). (**F–G**) Representative images used for quantification of color between CLNW observed (top) and explant imaged (bottom) within the same mouse. (**H–I**) As in (**F–G**) but a separate mouse. (**J–K**) As in (**F–G**) but a separate mouse.
DOI: https://doi.org/10.7554/eLife.33051.012

The following source data is available for figure 4:

**Source data 1.** Comparison of CLWN and explant derived data to calculate relative frequency, dominance, divergence index.
DOI: https://doi.org/10.7554/eLife.33051.013

## Single GC evolution in response to foreign antigen

To analyze single GC dynamics in response to foreign antigen, Confetti mice were immunized subcutaneously with NP-CGG, were tamoxifen treated, then received CLNW implants, and were followed over time (*Figure 5A*). For each time point, the frequency of the most dominant clone (*Figure 5B*) and the divergence index (*Figure 5C*) for every GC was quantified. The overall evolution towards pauciclonality was also quantifiable using a third metric, which we coined the meander index, and which is for all practical purposes the integral of the modulus for derivative of the divergence index, and could thereby also be considered a measure of distance traveled (*Figure 5D*). In agreement with previous observations, GC were found to evolve towards pauciclonality at widely varying rates (*Tas et al., 2016*). However, whereas the earlier work arrived at this conclusion based on population averages and variation within these averages across timed cohorts, the CLNW allowed resolution of individual GC over time, revealing additional information (examples in *Figure 5E–H*, please note that the representative pictures do not necessarily fully reflect the quantification plots, as the latter are based on several z-slices throughout the depth of the GCs). One GC, GC H, displayed a prominent double clonal inversion event (*Figure 5H*). GC H was found already by day seven to be >60% single-colored (RFP), indicating early clonal expansion. The expanded RFP clone(s) subsequently stagnated, dropped by day 13, and was supplanted by a YFP clone(s), which in turn was supplanted by a YFP+ mCFP clone(s) by day 19. Although one color does not necessarily signify a single clone in the present setup, and we therefore cannot exclude that one or several of the clonal events observed were driven by multiple clones, it appears highly unlikely that two consecutive clonal inversion events within the same GC were caused by such chance. Furthermore, prominent clonal inversion was also observed in GC F and G (*Figure 5F and G*), supporting the validity of the observation. One potential concern could be a phototoxicity issue, whereby the imaging itself could influence the predominant clones adversely, leading to outgrowth of minor clones. However, in agreement with previous observations, despite great heterogeneity, on average GCs were seen to evolve steadily towards pauciclonality at a rate comparable to that previously reported in time-point analyses (*Tas et al., 2016*), suggesting that clonal evolution was unperturbed in the present setup. In summary, the single-GC temporal resolution unmasked great heterogeneity in the evolution of GC over time. Our observations revealed that clonal inversion, although a rare event, does occur, and reveals a much greater GC fluidity than previously appreciated, underscoring the power of the longitudinal approach.

## Single GC evolution in an autoimmune setting

We recently developed a novel chimeric model displaying spontaneous, autoreactive GCs (*Degn et al., 2017*). Time-point analyses indicated that the autoreactive GCs in this model evolved towards pauciclonality at a similar rate as foreign antigen GCs. Using the CLNW, we reevaluated the kinetics using the longitudinal analysis (*Figure 6*). Mixed bone marrow chimeras were set up, and following reconstitution, chimeric mice were tamoxifen treated to switch on the Confetti reporter, and then received CLNW implants (*Figure 6A*). Again, the frequency of the most dominant color

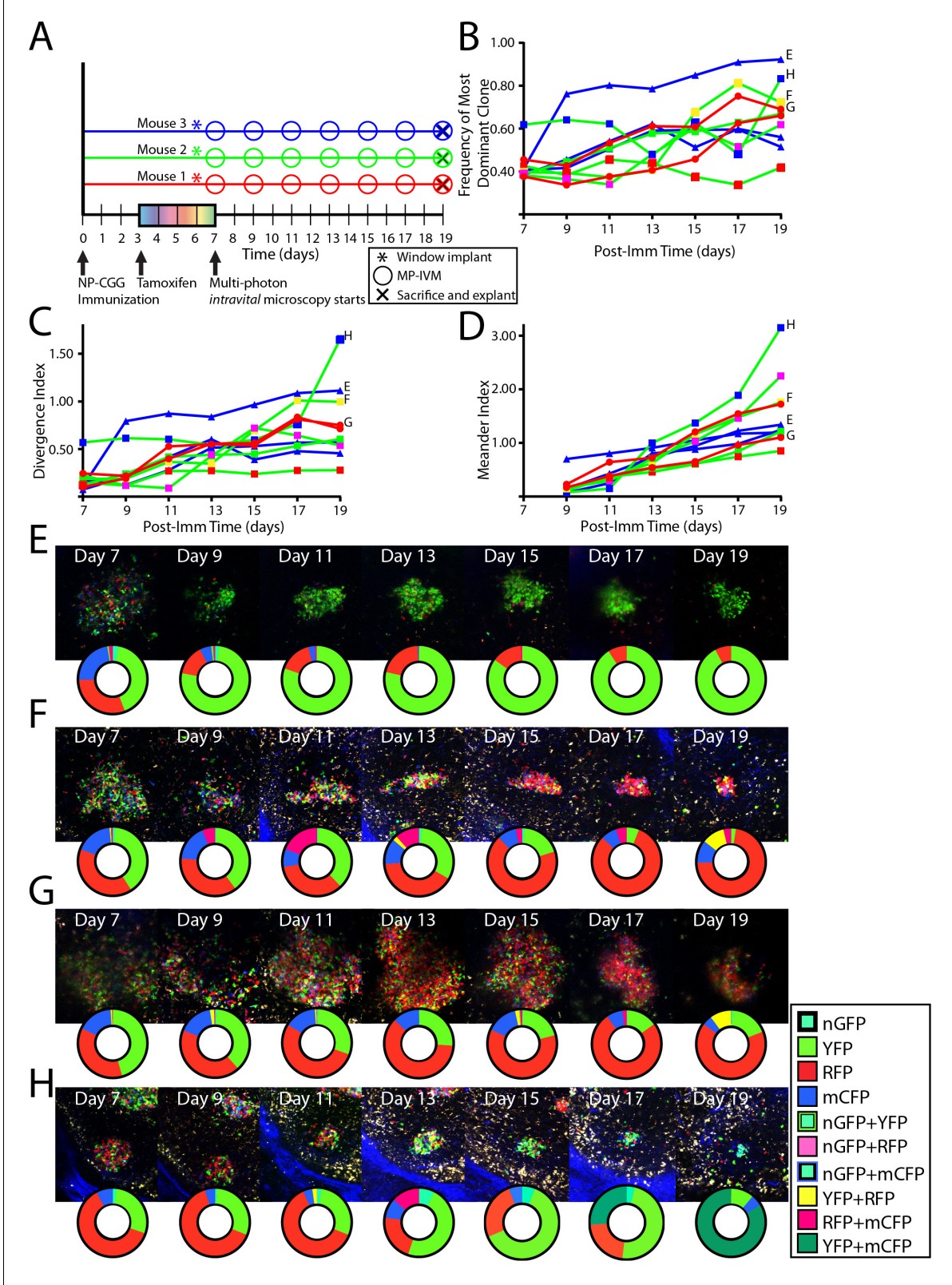

**Figure 5.** Following the clonal composition of individual GC responding to foreign antigen over days to weeks. (**A**) Schematic overview of experimental approach for NP-CGG immunization (imm), tamoxifen treatment, surgical implantation of the CLNW, and multiphoton intravital microscopy in CD45.2 *Aicda*-Cre-Confetti mice. Data represent one independent experiment with three mice each. (**B**) Frequency of most dominant color for each individual GC as observed from 7 to 19 days post-imm in 3 mice. (**C**) Divergence Index for each individual GC as observed from 7 to 19 days post-imm in 3 mice. *Figure 5 continued on next page*

*Figure 5 continued*

(D) Meander Index for each individual GC as observed from 7 to 19 days post-imm in 3 mice. (E) GC labeled E as in (B), (C), and (D) with pie chart to elaborate quantification below at each observation from days 7–19. (F) GC labeled F as in (B), (C), and (D) with pie chart to elaborate quantification below at each observation from days 7–19. Prominent clonal inversion observed from days 11–15. (G) GC labeled G as in (B), (C), and (D) with pie chart to elaborate quantification below at each observation from days 7–19. (H) GC labeled H as in (B), (C), and (D) with pie chart to elaborate quantification below at each observation from days 7–19. Double clonal inversion observed from days 11–13 and days 17–19. Please note that in E-H, representative images and quantification charts do not display 1:1 correspondence, since the latter are based on multiple z-planes throughout the volume of each of the GCs.

DOI: https://doi.org/10.7554/eLife.33051.014

The following source data is available for figure 5:

**Source data 1.** NPCGG tracking experimental data for each observed GC to calculate relative frequency, dominance, relative frequency, and meander score.

DOI: https://doi.org/10.7554/eLife.33051.015

(*Figure 6B*), the divergence index (*Figure 6C*), and the meander index (*Figure 6D*) were determined, and representative examples are shown in *Figure 6E* (again, representative images and quantification plots do not necessarily fully correlate, as the latter are based on multiple z-slices through the depth of the GCs). The findings in the autoreactive GC setting recapitulated those in the foreign-antigen scenario, and confirmed the possibility of clonal inversion (e.g., RFP clone(s) in *Figure 6E*, second row).

We additionally made the peculiar observation of a 'satellite cluster' of YFP cells outside an almost completely homogenized (RFP) GC by day 22 (third row, *Figure 6E*, see white arrow), followed by outgrowth of YFP cells in this GC. Whether these were residual YFP+ cells, regaining traction in the GC, or whether they were clones entering from outside this GC, was unclear. In several instances, we also observed that clonal bursts within one GC were followed by emergence of clones of similar color in neighboring GCs, raising the possibility of reentry of output memory B cells.

## Modeling GCs as ecological communities

Data analysis revealed similar GC dynamics for both NP-CGG immunized mice and mice reacting to self-antigen. However, we made the anecdotal observation that several adjacent GCs within the same animals had similar color composition or shifts in color composition later in the response; greater than would be expected to occur at random. Numerous prior studies have indicated that GC B cells do not traffic between GC. However, it has been demonstrated that output memory B cells can reenter GCs and rediversify. Thus, we sought to test whether there was evidence of synchrony between GCs using classical methods from ecology. In principal components analysis (PCA), large datasets with complex variable sets can be deconstructed to dimensions of variance. These 'principal components' where observations are stripped of their relatedness to x variable, but rather vectors of the covariance matrix, allow an exploration of population dynamics, since it provides insight into the internal structure of the data. It is important to note that it approaches the question of variance rather than a specific hypothesis and as such is exploratory in nature. Following principal components development, data were examined first visually and then statistically for clustering by mouse along the variant axes (*Figure 7A–C*). There were several clusters of GC observations, which were clearly distinguishable from the larger dataset and which turned out to be derived from three of the nine mice, termed A, B, and C. Following this discovery, the GCs observed within these mice were plotted individually to examine the trends in the dominant colors by time (*Figure 7D–F*). Whereas Mouse A demonstrated RFP+ dominance with increasing penetration across all three GC, Mouse B was striking for a contemporaneous spike in the relative abundance of CFP+ cells. Finally Mouse C appeared to have similar composition of largely YFP+ and RFP+ cells in both GCs. Upon regression analysis, as factor variables, each of Mouse A, B, and C were statistically significantly associated with PC1, PC2, and PC3; despite adjustment for time and color confounders. Although these data do not provide a direct observation of synchrony between GCs, they support statistically the observation that within the same animal, in different GCs, similar colors had a tendency to move together contemporaneously, suggesting the possibility of reentry of output memory B cell clones to ongoing GCs.

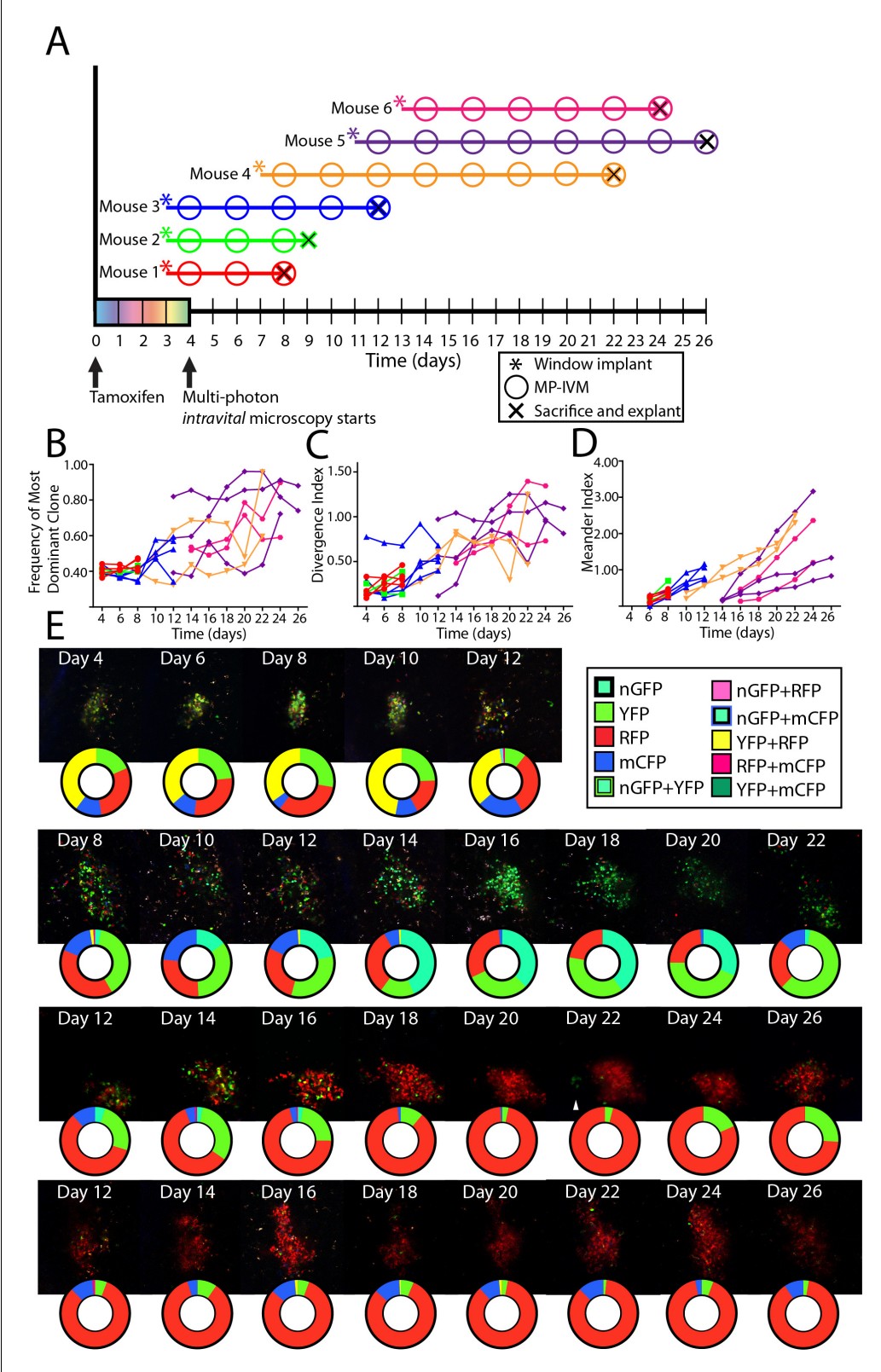

**Figure 6.** Following the clonal composition of individual autoimmune GC over days to weeks. (**A**) Schematic overview of experimental approach for CD45.2 *Aicda*-Cre-Confetti 564 mixed BM chimeras elaborating tamoxifen treatment, surgical implantation of the CLNW, and multiphoton intravital microscopy. Data represent two independent experiments with three mice each. (**B**) Frequency of most dominant color for each individual GC as observed from 4 to 26 days post-tamoxifen induction in individual GC from 6 mice. (**C**) Divergence Index for each individual GC as observed from 4 to

*Figure 6 continued on next page*

*Figure 6 continued*

26 days post-tamoxifen in 6 mice. (D) Meander Index for each individual GC as observed from 4 to 26 days post-tamoxifen in 6 mice. (E) Four representative GC labeled with pie chart to elaborate quantification below at each observation from days 4–26. Second row down, prominent double inversion event comparing days 12–14 and 20–22. Third row down, white arrow for marking the site of a cluster of extra-GC YFP+ cluster which compete in the RFP +dominant GC on days 24 and 26. Please note that the representative images and quantification charts do not display 1:1 correspondence, since the latter are based on multiple z-planes throughout the volume of each of the GCs.

DOI: https://doi.org/10.7554/eLife.33051.016

The following source data is available for figure 6:

**Source data 1.** 564-Ig tracking experimental data for each observed GC to calculate relative frequency, dominance, relative frequency, and meander score.

DOI: https://doi.org/10.7554/eLife.33051.017

## Discussion

Our development of the present protocol has been prevised by several elegant studies. Based on static and dynamic imaging data, Jenkins and colleagues generated a numerically, spatially, and temporally scaled simulation of the first 50 hr of the primary T cell-dependent immune response (*Catron et al., 2004*). Ito et al. employed an inguinal lymph node window chamber to conduct serial, but discontinuous, imaging of cancer cell dissemination in vivo (*Ito et al., 2012*). Padera and colleagues developed a chronic lymph node window model to investigate the growth and spread of lymph node metastases (*Jeong et al., 2015*; *Meijer et al., 2017*). Here we present longitudinal B cell imaging covering the first 150 hr of the primary T cell-dependent immune response to foreign antigen, from initial formation of primary foci to seeding of follicular GCs. We followed the clonal evolution of foreign antigen elicited GCs for up to three weeks; and we have similarly followed the clonal evolution of autoreactive GCs on a three-week time frame.

Follow-up of early time events has previously been hard to achieve, as classical MP-IVM approaches have been limited to time-frames of 8–12 hr maximum. While our results were overall in agreement with prior findings and with the simulation of Catron et al (*Catron et al., 2004*)., some additional insights could be gleaned from the longitudinal time resolution. First and not surprisingly, high-affinity B cells specific for the immunized antigen appeared in the paracortical region around 72 hr after immunization and then over the course of the next 12–24 hr migrated into the follicle where GCs developed. Whereas two of eight observed GCs (Fo1 and 4) demonstrated a neat exponential growth phenomenon with doubling time closely approximating the shorter end of past work using mathematical modeling following BrdU labeling (*Table 1*, *Figure 3—video 1* and *Figure 3—video 2* and [*Anderson et al., 2009*]), a further three demonstrated lower but still reasonable doubling time around ~12 hr (Fo2, 6, and 7) and the final three demonstrated longer doubling times > 12 hr (Fo3, 5, and 8). In silico analysis of the early response has previously been performed using average rates of division and cell death across multiple explanted GCs for fixed low and high affinity clones (*Anderson et al., 2009*). In contrast, this study relies on the affinity of the B1-8hi clone in the GC to estimate dynamics. Specifically, in GCs where the B1-8hi clones dominate, the doubling time observed will most closely approximate the 'true' underlying doubling rate as more of the GC is observed. However, in GCs with more competitive clones that are dark, the observed rate may deviate from the true GC dynamics since a greater proportion is unobserved. As competitive factors and/or GC viability factors begin to affect the dominance of the CFP+ clones, the estimates of doubling time will be artificially elevated, that is slower, as it has been shown that cell death as opposed to cell division is the primary effector of affinity-driven selection (*Anderson et al., 2009*). Thus GCs where the CFP+ clones are dying at relatively greater rates will have estimates with slower doubling times since the rest of the GC is unmeasured. Despite this caveat, within the same lymph node and within the same animal subjected to timed antigen exposure, expansion of primary foci and seeding of GCs, even in adjacent follicles, was found to occur at highly variable rates (*Table 1*).

However, several GCs that appeared to have adequate 'seeding' (by comparing relative infiltration of CFP+ cells) later went dark. This black-out phenomenon was observed in three of eight Fo (Fo3, 5, 8), which is surprising due to the significantly greater baseline affinity of the B1-8hi clones compared to the resting repertoire and may indicate (i) failure to establish a productive GC, (ii) entry of an endogenous dark clone with similar or superior affinity for the antigen and subsequent

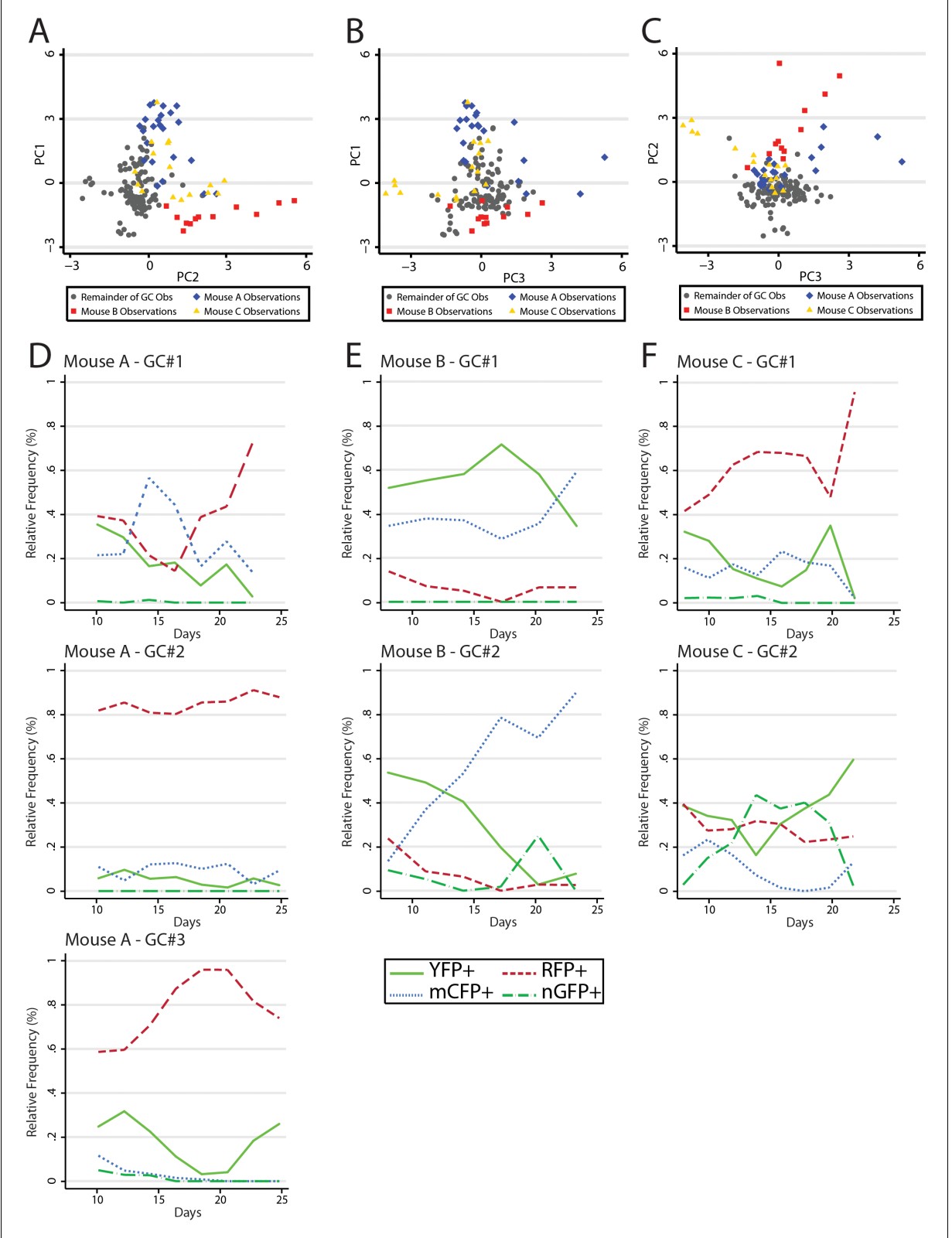

**Figure 7.** Principal component analysis reveals global mouse association with variance in color relative frequency for adjacent GC. (**A**) Following generation of principal components, GC data points are shown by principal component 1 (PC1) and principal component 2 (PC2). Data points are colored as Mouse A (blue), Mouse B (red), Mouse C (gold), and the remainder of GC (gray). Statistical association for PC1 was p<0.0001 for A, p<0.0001 for B, and p=0.5861 for C. For PC2 A was p=0.142, B was p<0.0001, and C was p<0.0001. For PC3 A was p=0.002, B was was p=0.005 and C was

*Figure 7 continued on next page*

*Figure 7 continued*

p=0.002. (**B**) As in (**A**) but for PC1 and PC3. (**C**) As in (**A**) but for PC2 and PC3. (**D**) Relative frequencies of the four most prevalent colors in all observed GC (#1–3) within the window of Mouse A. (**E**) As in (**D**) but for Mouse B. (**F**) As in (**D**) but for Mouse C.

DOI: https://doi.org/10.7554/eLife.33051.018

The following source data is available for figure 7:

**Source data 1.** Longitudinal observations of relative frequency by GC to perform PCA

DOI: https://doi.org/10.7554/eLife.33051.019

'winning-out' of the GC leading to loss of CFP+, or (iii) that early competition took place and SHM led to the generation of a dark clone with greater affinity than the B1-8hi clones adoptively transferred prior to immunization. Although more difficult to assess due to complex technical limitations, (excitation wavelengths and spectral overlap of multiple fluorochromes with the multiphoton microscopy approach as well as genetic limitations) access to FDCs and to T-cell help may play a role in the described variation. Nonetheless this observation in multiple GCs followed across time in separate animals deserves further exploration.

Combining our intravital imaging approach with the *Aicda*-Confetti reporter of Tas et al., the process of clonal evolution could be followed over time with single-GC resolution. Again, our overall findings were in agreement with previous observations, but the longitudinal resolution allowed for additional layers of detail to be uncovered. The average clonal evolution towards pauciclonality was found to mask complex underlying processes entailing both clonal bursts, as previously reported, but also clonal 'inversion events', whereby a seemingly winning color was suddenly seen to lose competitive momentum allowing for emergence of one or several new colors. In the case of a single-color take-over, this event could of course be reflective of a clonal burst occurring in the new winning color, whereas this explanation fails to account for multi-color take-overs, since simultaneous clonal bursts are unlikely. Additionally, we cannot account for a drift in antigenic reactivities over time. It was recently reported that complex antigens drive permissive clonal selection in GCs (*Kuraoka et al., 2016*). Although we employed NP-CGG immunizations, where NP would be expected to be the immunodominant epitope on the B cell side, the CGG is a more complex mixture of chicken gamma-globulins, which may cater to multiple layers of T follicular helper cells, which could potentially skew responses over time. In this context, it is important to note that chiefly B cells responding to the same antigen or catering to the same T helper cells, would be able to enter an ongoing GC (*Schwickert et al., 2007*; *Schwickert et al., 2009*). This scenario is vastly more complex in the ARTEMIS model, which involves a very broad autoreactive repertoire, with increasing epitope spreading over time (*Degn et al., 2017*).

Of note, the GC that displayed the prominent sequential dual clonal inversion event, GC H (*Figure 5H*), was at the outset relatively small, and the final take-over by a YFP +mCFP clone coincided with a decrease in GC size coincident with the latest stages of the GC response. It is unclear what determines the size of GCs, and the notable variability in GC size within for example a single lymph node of an individual. One could speculate that it is related to the degree of successful affinity maturation, and in turn that clonal inversion is a more likely event in smaller, less successful, GC. This is supported by studies in both mice and humans that demonstrate larger GC structures in AID deficient subjects who have less SHM and affinity maturation (*Fagarasan et al., 2002*; *Revy et al., 2000*). However, this supposition is contraindicated in the present dataset by the clonal inversion event observed in GC G (*Figure 5G*), which was a massive GC.

Conversely, one could speculate that GCs that homogenize quickly do not 'travel far', and may be suggestive of less affinity matured clones as compared to GC which 'churn' through many clones with mutations accumulating on the way to clonality. However, elucidating this question would require an additional in-depth analysis of the mutational landscape across the observed clonal populations.

Finally, our studies were extended to the recently described novel ARTEMIS model for spontaneous autoreactive GCs (*Degn et al., 2017*) in combination with the Confetti reporter. This revealed that autoreactive GC responses were largely reflective of foreign antigen elicited GC clonal dynamics, and confirmed our insights from the foreign antigen GC scenario in an independent model.

It has previously been reported that GCs are open and dynamic structures (*Schwickert et al., 2007*), and that newly activated B cells can enter ongoing GCs (*Schwickert et al., 2007*). Whether this extends to short-term memory cells, allowing reseeding of GCs in a chronic GC setting remains an open question. However, secondary remodeling of BCRs of memory B cells has been observed (*McHeyzer-Williams et al., 2015*). In several instances, we observed that clonal bursts within one GC were followed by emergence of clones of similar color in neighboring GCs, suggestive of some level of inter-GC communication at the B cell level (*Figure 7*). Integrating the above-mentioned considerations derived from the available literature, this would not occur directly at the GC B cell level, but rather by exit and subsequent reentry of (short-term) memory B cells. At the time-scales presented here, this would only be expected to occur due to high (auto)antigen levels driving immediate re-engagement of the BCR following GC exit. Although this remains at present inference, future studies employing an *Aicda*-driven reporter coupled with CD40L-blockade to ablate all ongoing GCs and a subsequent investigation of reporter B cell reentry in reemergent GCs could provide greater clarity.

Along the same lines, our finding of trafficking between the high-density CFP+ core of dividing cells in the GC and the paracortical clusters (presumably primary foci) between 72–120 hr after immunization (*Figure 3—video 1*), may demonstrate a mechanism by which early, pre-GC, competition may influence the clonal composition of the GC and may contribute to the 'black-out' phenomenon described above.

The validity of the CLNW model was ascertained in experiments comparing weight curves between sham-operated and window-implanted animals, as well as basic cellular parameters of secondary lymphoid tissues, including window-imaged and contralateral inguinal lymph nodes. The most notable differences between sham and window animals were the increase in Gr1 cells in the spleen and the slight, but statistically significant increase in GC B cells in the window node (*Figure 2*). The latter finding in particular could be cause for concern in relation to the investigations of clonal dynamics presented here. Two factors could contribute to these differences: 1) the sterile, local inflammation driven by disruption of the overlying tissue, however gentle, and exposure of the underlying tissues; and 2) a non-sterile inflammation driven by exposure of the chamber to ambient skin commensals and opportunistic pathogens. Naturally, the former cannot completely be avoided, whereas the latter would be a greater cause for concern. We employed broad spectrum perioperative antibiotics in both sham and CLNW animals an effort to reduce the risk of meaningful perturbation further. Additionally, the comparison of clonal evolution in window and contralateral lymph nodes within the same animals, presented in *Figure 4*, confirmed that this process was not affected by the CLNW. Any potential effect might in part also be alleviated by the experimental setup, whereby immunization (foreign antigen GC) or establishment of autoreactive GCs in the chimeras, and tamoxifen-induced activation of the reporter precedes surgical placement of the window. Therefore, any potential GC response associated with the surgical procedure or subsequent contamination of the chamber (although none such were noted by us in the experiments presented), would be driven by unlabeled clones, which would not directly perturb results. Indirectly, outgrowth of a significant body of unlabeled, 'dark' clones, targeting ambient antigen (rather than the immunogen NP-CGG, or autoantigen in the autoreactive GC) would instead lead to GCs going dark. However, we did not observe any GCs going dark for the duration of the imaging experiments presented here, and the GCs that were followed displayed a consistent and robust labeling density, indicating that this was not an issue in the presented setup.

Therefore, with the present setup, cellular immune responses can now be followed at a temporal resolution of hours and days and weeks per experimental question at hand.

## Materials and methods

**Key resources table**

| Reagent type (species) or resource | Designation | Source or reference | Identifiers | Additional information |
| --- | --- | --- | --- | --- |
| Antibody | Rabbit polyclonal anti-B-Phycoerythrin | Rockland Immunochemicals | Cat#200-401-099; RRID: AB_10893993 | |

*Continued on next page*

*Continued*

| Reagent type (species) or resource | Designation | Source or reference | Identifiers | Additional information |
|---|---|---|---|---|
| Antibody | Rabbit polyclonal anti-C3b-A633 | This paper | N/A | |
| Antibody | Rat monoclonal anti-mouse/human GL7 antigen-PacBlue (clone GL7) | Biolegend | Cat#144613; RRID: AB_2563291 | |
| Antibody | Rat monoclonal anti-mouse /human CD45R/B220-PerCP/Cy5.5 (clone RA3-6B2) | Biolegend | Cat#103235; RRID: AB_893356 | |
| Antibody | Mouse monoclonal anti-mouse IgM$^b$-FITC (clone AF6-78) | Biolegend | Cat#406205; RRID: AB_315038 | |
| Antibody | Mouse monoclonal anti-mouse IgM$^a$-PE (clone MA-69) | Biolegend | Cat#408608; RRID: AB_940545 | |
| Antibody | Mouse monoclonal anti-mouse CD45.1-FITC (clone A20) | Biolegend | Cat#110705; RRID: AB_313494 | |
| Antibody | Mouse monoclonal anti-mouse CD45.2-APC (clone 104) | Biolegend | Cat#109813; RRID: AB_389210 | |
| Antibody | Rat monoclonal anti-mouse IgD-PacBlue (clone 11–26 c.2a) | Biolegend | Cat#405711; RRID: AB_1937245 | |
| Antibody | Rat monoclonal anti-mouse CD21/CD35 (CR2/CR1)-PE (clone 7E9) | Biolegend | Cat#123409; RRID: AB_940411 | |
| Antibody | Rat monoclonal anti-mouse CD138 (Syndecan-1)-PE (clone 281–2) | Biolegend | Cat#142503; RRID: AB_10915989 | |
| Antibody | Rat monoclonal anti-mouse CD38-PE/Cy7 (clone 90) | Biolegend | Cat#102717; RRID: AB_2072892 | |
| Antibody | Rat anti-mouse CD31-A647 (clone 390) | Biolegend | Cat#102415 RRID: AB_493411 | |
| Antibody | Mouse monoclonal anti-mouse CD157 (BST-1)-PE (clone BP-3) | Biolegend | Cat#140203; RRID: AB_10643273 | |
| Antibody | Mouse monoclonal anti-mouse CD95 (APO-1/Fas)-PE (clone 15A7) | eBioscience | Cat#12-0951-81; RRID: AB_465788 | |
| Antibody | Rabbit polyclonal anti-Goat IgG (H + L) Cross-adsorbed-A488 | ThermoFisher Scientific | Cat#A-11078; RRID: AB_2534122 | |
| Antibody | Goat polyclonal anti-Mouse IgG$_{2c}$, Human adsorbed-AP | Southern Biotech | Cat#1079–04; RRID: AB_2692321 | |
| Antibody | Goat polyclonal anti-Mouse IgG$_{2a}$, Human adsorbed-AP | Southern Biotech | Cat#1080–04; RRID: AB_2692322 | |
| Antibody | Rabbit polyconal anti-Mouse Immunoglobulins-biotin | DAKO | Cat#E035401-2; RRID: AB_2722694 | |
| Antibody | Mouse monoclonal anti-idiotype (clone 9D11) | *Chatterjee et al. (2013)* | N/A | |
| Other | Custom Chamber | This paper | N/A | |
| Other | Microsurgical Instruments | Fine Science Tools, Foster City, California, USA | N/A | |
| Other | 5–0 ethilon and 5–0 stainless steel suture | Ethicon, Somerville, New Jersey, USA | N/A | |
| Chemical compound, drug | Tamoxifen | Sigma | Cat#T5648 | |

*Continued on next page*

*Continued*

| Reagent type (species) or resource | Designation | Source or reference | Identifiers | Additional information |
|---|---|---|---|---|
| Chemical compound, drug | Imject Alum Adjuvant | ThermoFisher Scientific | Cat#77161 | |
| Chemical compound, drug | Np-Osu | Bioresearch technologies | Cat#N-1010–100 | |
| Chemical compound, drug | Fixable Viability Dye eFluor 780 | eBioscience | Cat#65-0865-14 | |
| Chemical compound, drug | Hoechst 33342 | ThermoFisher Scientific | Cat#H3570 | |
| Chemical compound, drug | DAPI (4′,6-Diamidino-2-Phenylindole, Dihydrochloride) | ThermoFisher Scientific | Cat#D1306 | |
| Peptide, recombinant protein | B-Phycoerythrin | ThermoFisher Scientific | Cat#P800 | |
| Peptide, recombinant protein | Streptavidin-PE/Cy7 | Biolegend | Cat#405206 | |
| Peptide, recombinant Protein | Phalloidin-A568 | ThermoFisher Scientific | Cat#A12380 | |
| Peptide, recombinant protein | Chicken gamma globulin | Rockland Immunochemicals | Cat#D602-0100 | |
| Peptide, recombinant protein | Europium-labeled streptavidin | Perkin Elmer | Cat#1244–360 | |
| Biological sample (*Mus musculus*) | Mouse: *Aicda*-Cre[ERT2] EYFP: *Aicda*-CreERT2 flox-stop-flox-EYFP | *Dogan et al. (2009)* | N/A | |
| Biological sample (mouse) | Mouse: PA-GFP: B6.Cg-*Ptprc*[a] Tg(UBC-PA-GFP)1Mnz/J | The Jackson Laboratory | JAX: 022486 | |
| Biological sample (mouse) | Mouse: 564Igi: 564 HiKi | *Berland et al. (2006)* | N/A | |
| Biological sample (mouse) | Mouse: *Aicda*-Cre[ERT2] Confetti: *Aicda*-CreERT2-Rosa26Confetti | *Tas et al. (2016)* | N/A | |
| Software, algorithm | ImageJ | Rasband WS, ImageJ, NIH, Bethesda, Maryland, USA | https://imagej.nih.gov/ij/ | |
| Software, algorithm | STATA | Statacorp, College Station, Texas, USA | https://stata.com | |

## Mice and reagents

C57BL/6 were obtained from Jackson Labs (Bar Harbor, ME) and maintained in our SPF animal facility under the supervision of Harvard Medical School IACUC (protocol numbers IS00000095 and IS00000111). B1-8hi CFP donors for adoptive transfers were kindly provided by Gabriel Victora (Whitehead Institute, MIT, Cambridge). *Aicda*-CreERT2-Confetti mice were previously generated (*Tas et al., 2016*). These mice are heterozygous carriers of tamoxifen-regulated Cre under the control of the Aicda-gene, and harbor biallelic knock-in of a stop-floxed expression cassette containing 4 XFPs (mCFP, nGFP, cYFP and cRFP) flanked by mutually exclusive loxP sites. Expression of Cre in the presence of tamoxifen allows stochastic recombination of a single, or less frequently, both alleles, to express any one XFP or combination of two independent XFPs, yielding 10 possible color combinations. Cre activity ceases following the pharmacokinetic clearance of tamoxifen, somewhere between 3 and 4 days for a single oral gavage administration, and at this point recombined cells are locked in. Thus, the strain can be utilized in pulse-chase experiments where a temporal population of germinal center B cells (expressing AID) and all their future progeny can be committed to genetically hard-wired expression of one out of 10 possible XFP combinations. It is subsequently possible to track the clonal dynamics of GC B cells.

Quantification was performed using manual counting. Antibodies against CD45, Gr-1, CD4, CD8, B220, GL7, and CD38, were obtained from Biolegend (San Diego, CA) and used between 1/200-1/

500 dilutions from 1 mg/ml starting concentrations per our experience. Flow cytometry was performed using a FACS Canto II (for non-Confetti controls) or FACS ARIA SORP (for Confetti analysis and cell sorting) (BD Biosciences, San Jose, CA).

## Bone marrow chimeras

Recipients were irradiated with 950 rad, then immediately placed on water containing antibiotics (sulfamethoxazole/trimethoprim) to prevent opportunistic infections during the reconstitution phase. Femurs and tibia were extracted from bone marrow donors, mechanically cleaned and rinsed through several rounds of sterile filtered HBSS containing 10 mM HEPES, pH 7.2, 1 mM EDTA and 2% heat-inactivated FBS (BM buffer). The bones were subsequently crushed in a mortar, and the cell extract was passed through a 70 micrometer cell strainer (Corning). An aliquot was subjected to RBC lysis and counted in a standard hemacytometer (Neubauer chamber). Based on cell counts, appropriate ratios of mixed bone marrow were calculated to achieve final desired donor ratios. Cells were pelleted by centrifugation (200 $g$, 5 min) and resuspended at $1 * 10^8$ cells per ml, and 100 µl was injected i.v. into each irradiated recipient ~10–12 hr post irradiation.

## Adoptive transfers

B1-8hi CFP donor mice were sacrificed by isofluorane overdose followed by cervical dislocation, and spleens were harvested into ice-cold FACS buffer (PBS, 2% FCS, 1 mM EDTA), mechanically dissociated, then passed through a 70 µm cell strainer, followed by centrifugation at 200 $g$ for 5 min. The supernatant was decanted and the pellet resuspended in RBC lysis buffer (155 mM $NH_4Cl$, 12 mM $NaHCO_3$, 0.1 mM EDTA), incubated for 2–3 min then spun down as before, washed with PBS, and resuspended in FACS buffer containing a cocktail of antibodies for untouched B cell purification (biotinylated antibodies from BD Biosciences: CD11b, CD11c, CD69, CD3ε, TCRβ, Ter119). Cells were incubated with staining mix on ice for 30 min, then washed, and streptavidin magnetic beads (Pierce) were added in FACS buffer. After another 15 min incubation on ice, cells were again washed, then resuspended in FACS buffer and loaded on a prewetted MACS LD column. The flow-through containing untouched purified B cells was collected and cells were immediately transferred to recipients. Recipients were anaesthetized with isofluorane and cell suspensions were injected i.v. through the retroorbital sinus. Flow cytometric analysis of sample cell suspensions estimated ~50,000 B1-8hi CFP + cells per transfer.

## Immunizations

A hapten immunization scheme utilizing 4-hydroxy-3-nitrophenylacetyl-chicken gamma globulin (NP-CGG) was chosen to mimic responses to a model foreign antigen. Immunization was conducted peripherally at the hock and groin bilaterally (20 µg/injection) in equal volume alum adjuvant. Immunization was conducted 3 days prior to tamoxifen administration to generate primitive GCs specific for NP-CGG prior to labeling the GC compartment. Past work in our lab has established a recombination efficiency between 45% and 50% using a single tamoxifen gavage. Mice were longitudinally imaged following implantation of a surgical chamber (as described below) or were sacrificed at various time points following NP-CGG immunization (explant analysis) and GC read out by multiphoton microscopy, FACS, or imaging cytometry.

## Surgical chambers and supplies

Surgical chambers were developed previously (*Jeong et al., 2015*) to follow tumor metastasis and custom milled for our purposes. They consist of a top and bottom plate of titanium containing a circular cut out for placement of an 11.7 mm circumference coverslip (*Figure 1*). These two plates are superimposed in a mirror configuration with a section of skin brought to the lateral border of the plates which is then sutured in place. Surgical supplies were those used for basic microsurgical technique, fine microsurgical instruments (Fine Science Tools, Foster City, CA), 5–0 ethilon and 5–0 stainless steel sutures (Ethicon, Somerville, NJ).

## Microsurgical technique

Mice were used around 8 weeks of age. Although use of older mice was technically feasible, the age-associated increase in subcutaneous adipose tissue was found to negatively impact the process

of locating the node with minimal dissection. Conversely, the use of younger mice was challenging due to their smaller size, rendering window implantation risky due to excessive skin stretching and limiting chamber lifespan. Instruments and window implant components were washed to remove any macroscopic contamination and then autoclaved to yield sterile surgical tools. Prior to surgery, both sham and CLNW implanted mice received one-week perioperative broad spectrum antibiotics, sulfa-methoxazole-trimethoprim, in their drinking water (SMZ/TMP oral suspension, Hi-Tech Pharmacal, Amityville NY). Mice were prepared using standard surgical technique, the area to receive the window implant was shaved (approximately from the right costal margin down to the inguinal fossa medially and iliac crest laterally) and then prepped using alternating washes of ethanol and betadine. Following site preparation and draping, the surgical instruments were opened in a sterile fashion and the surgeon donned sterile gloves. Next the inguinal/subiliac LN was palpated, typically ~4–6 millimeters cranial to the iliac crest and ~3 millimeters lateral to the fifth or inferior inguinal nipple. The chamber to be implanted was briefly overlaid over the site to visualize the location of the node and necessary incisions for the anchor points, then removed from the field again. Incisions were made cranially and caudally to allow for one screw anchor at each location. The top plate with two screw anchors inserted was placed into the incision, and the screw anchors were carefully passed through the connective tissue underlying the dermis and out on the other side where they were gently tightened into the corresponding holes of the bottom plate. Upon correct placement, skin was seen to extend >75% of the way across the interior cut out area without any manipulation, as less was found to result in excessive stretching leading to early failure. Skin anchors were placed with 5–0 sterile suture; using ethilon for shorter duration and stainless steel for longer duration of the chamber. Three skin anchors were secured at the lateral margin, bringing skin across the entire interior of the chamber. Using a surgical stereoscope with 10-25X magnification, an incision was made above the location nearest approximating the LN through the epidermis only, taking care to avoid the underlying dermal fascia containing small vessels. After incision, several avascular planes were appreciable through which the subdermal fascia could be entered; a plane was chosen that was parallel to the direction of the LN and the space was entered using blunt dissection. The fascia was gently retracted on each side and the incision was extended towards the LN. Once the node was visualized, 2–3 layers of connective tissue overlying the perilymphatic adipose tissue and then a single layer just superior to the capsule was gently taken down, in order to expose a few millimeters of the node. The exposed LN was kept irrigated to prevent it from desiccating until the chamber was closed with a coverslip. The coverslip was placed over a pool of sterile irrigation overlying the node and gently pushed to force this fluid out, creating a vacuum. A flexible, stainless steel O-ring was placed into the groove securing the coverslip on the top plate. Finally, stay sutures were placed on the medial anchor points on both the top and bottom plates. A coverslip and O-ring was placed on the back plate chamber to protect the exposed posterior skin and to prevent exterior contamination of the environment inside. Mice received typical buprenorphine s.c. injections perioperatively for the first 48 hr. CLNW implanted mice were noted to remain less mobile for the first 24–48 hr post-op than sham controls, however, they were remarkably agile thereafter with minimal, if any, restrictions due to pain or discomfort (including climbing upside down in the cages). Notably, mice were single-housed post-operatively as is typical.

## Multiphoton microscopy

Multiphoton intravital microscopy (MP-IVM) was utilized to visualize fluorescently labeled cells within the lymph node. All imaging was performed on an upright Olympus FV1200 MPE multiphoton system microscope fitted with either a 20 × 0.95 NA Plan water-immersion objective or a 25 × 1.05 NA Plan IR optimized water-immersion objective, a MaiTai HP DeepSee Ti-Sapphire laser (Spectraphysics), and four non-descanned detectors (2 GaAsP and two regular PMTs). Imaging of Confetti alleles was performed using λ = 940 nm excitation. Fluorescence emission was collected in three channels, using the following filter sets: a pair of CFP (480/40 nm) and YFP (525/50 nm) filters, separated by a 505 nm dichroic mirror, for CFP/GFP/YFP detection, and a third filter (605/70 nm) for RFP detection. Prior to imaging, the mice were anesthetized using an isofluorane vaporizer; with 1.5–2% isofluorane for induction and 0.5–1% for maintenance. Animals were maintained on an $H_2O$ reservoir which was kept at ~34 degC to maintain body temperature. Following induction of a stable plane of anesthesia, mice were placed into a custom-built Plexiglas fixture and the chamber grasped firmly by fixable alligator grips to stabilize the imaging plane. The microscope was centered on the node and focused

on the top of the node using transmitted light and then convert to dark room operations for imaging. Mice were continuously monitored during imaging for vital signs (respiratory rate and pulse) and adequate plane of anesthesia (toe pinch reflex). Imaging was performed either daily or on alternating days using 640 × 640 pixel resolution without Kalman line correction to avoid phototoxicity.

## Statistics and quantitative analysis

Flow cytometry data was quantified using FlowJo (ver. 8, Ashland, OR). BM chimeras using the Confetti allele were quantified manually following acquisition. Observers were blinded to animal status and interobserver reliability confirmed by independent counts of 15 GC with concordance >95% between two observers. Counting was performed on 3–4 z planes from stacks acquired at 5 µm steps through the GC. Slices were at least 15–20 µm apart to preclude possibility of double counting. Very sparse, <50% occupation GC were discarded prior to analysis as they are considered to have at least one expanded dark clone. Typically, greater than 250 GC B cells were quantified for a single GC per time point. Raw counts were converted to relative frequencies for each color possibility to simplify analysis. Clonal dominance was calculated as the frequency of the most dominant clone/color at a given time point. The meander score is mathematically the integral of the relative velocities for each color shown below:

$$Meander\,score(t) = \int_0^t \sum_{i=1}^{10} \sqrt{\left(freq_{color(i)_{(t)}} - freq_{color(i)_{(t-1)}}\right)^2}$$

Clonal divergence score was reported previously (*Tas et al., 2016*) and is calculated at a given time point by comparing the observed distribution of colors at time t to the expected frequencies if the distribution was randomly recombined. Expected frequencies for each of 10 possible colors was derived using explant data from day three post tamoxifen treatment where color distribution most closely mimics random recombination and prior to any selection events:

$$Clonal\,divergence\,score\,(t) = \sum_{i=1}^{10} |observed\,freq_{color_{(i)}} - expected\,freq_{color_{(i)}}|$$

Descriptors of GC behavior and clonality were graphed using GraphPad Prism 6 (GraphPad Software Inc, La Jolla, CA). Statistical inference was tested using chi squared or one-way or two-way ANOVA as appropriate depending on the number of groups. Principal components analysis (PCA) was performed in the standard fashion and resulting dimensions tested for association with individual mice after adjusting for time and treatment. All observations were repeatedly in at least two independent experiments and which is indicated in the respective presentation of the data. As this was an exploratory analysis of longitudinal variation we did not perform sample size calculations in advance of the study design. Group sample sizes were based on technical and biological variation observed in prior studies and emphasis was placed on confidence intervals when interpreting variance to give better contextualization of all data points. All statistical inference utilized Stata version 13 (StataCorp, College Station, TX).

## Acknowledgements

We are grateful to Luka Mesin and Gabriel Victora from Rockefeller University for the B1-8hi CFP and confetti allele animals and for related technical consultation and Eelco Meijer from the MGH for the window chambers. We finally thank Harry Leung of the Optical Microscopy Core at the PCMM for technical assistance. DJF was supported by a Howard Hughes Medical Institute, Medical Research Fellowship. SED was supported by the Benzon Foundation and by a Marie Curie International Outgoing Fellowship within the seventh European Community Framework Programme. This research was supported by NIH (R21AI117737, R01AI039246) (MCC), and the Alliance for Lupus Research (MCC).

## Additional information

### Funding

| Funder | Grant reference number | Author |
| --- | --- | --- |
| Howard Hughes Medical Institute | | Daniel J Firl |
| Marie Curie International Fellowships | | Soren E Degn |
| Alfred Benzon Foundation | | Soren E Degn |
| National Institutes of Health | R21AI117737 | Michael C Carroll |
| National Institutes of Health | R01AI039246 | Michael C Carroll |
| Alliance for Lupus Research | | Michael C Carroll |

The funders had no role in study design, data collection and interpretation, or the decision to submit the work for publication.

### Author contributions

Daniel J Firl, Conceptualization, Data curation, Formal analysis, Validation, Investigation, Visualization, Methodology, Writing—original draft, Writing—review and editing; Soren E Degn, Conceptualization, Supervision, Investigation, Visualization, Methodology, Writing—original draft, Writing—review and editing; Timothy Padera, Resources, Writing—review and editing; Michael C Carroll, Resources, Supervision, Funding acquisition, Project administration, Writing—review and editing

### Author ORCIDs

Daniel J Firl (iD) http://orcid.org/0000-0001-7993-6167
Soren E Degn (iD) http://orcid.org/0000-0001-5409-045X
Michael C Carroll (iD) http://orcid.org/0000-0002-3213-4295

### Ethics

Animal experimentation: Animal studies supervised by the Institutional Animal Care and Use Committee of Harvard Medical School (protocol numbers IS00000095 and IS00000111) as described in the manuscript.

### Decision letter and Author response

Decision letter https://doi.org/10.7554/eLife.33051.023
Author response https://doi.org/10.7554/eLife.33051.024

## Additional files

### Supplementary files

• Transparent reporting form
DOI: https://doi.org/10.7554/eLife.33051.021

### Data availability

All data generated or analysed during this study are included in the manuscript and supporting files.

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
