## [Decision Letter]

Thank you for submitting your article "Capturing Change in Clonal Composition amongst Single Germinal Centers" for consideration by *eLife*. Your article has been reviewed by three peer reviewers, including Michael L Dustin as the Reviewing Editor and Reviewer #1, and the evaluation has been overseen Michel Nussenzweig as the Senior Editor.

The reviewers have discussed the reviews with one another and the Reviewing Editor has drafted this decision to help you prepare a revised submission.

Summary:

Your manuscript presents a modification of an imaging preparation to permit longitudinal studies in lymph nodes. The authors use it to address imaging of germinal centers over the time course of three weeks. This is a helpful development, with many potential applications in immunology.

They demonstrate that the clonal populations in GCs are more dynamic and heterogeneous in their evolution than had previously been reported by imaging of individual time points, and that GCs can "devolve" – i.e., a highly represented clone can be replaced by another clone. This finding is important and interesting, but "devolution" is not the only way to conceptualize this and based on the level of evidence presented in this tools/resources article, other interpretations should be discussed and if the method could eventually resolve different possibilities. There is also some concern about the scope of evidence that preparation is not altering the function of the lymph node and the rigour with which results and reported and discussed, which should be addressed prior to publication. The revision may present additional data to address some points, but more generally the revision should address, as quantitatively as possible, the potential of the method to address important issues raised in a definitive manner. The highest priority is to make sure the tool/resource described is suitable to definitively address some of the issues raised by the initial results collected while validating the technical approach.

Essential revisions:

1) One of the most important applications of the method is the idea that B cell clones may move between GC, which is thought not to take place. You note that in some cases neighbouring GCs exhibit similar color changes at the same time and suggest that this may be due to GC trafficking. The proposed mechanism is unclear – do the authors have evidence that GC cells are found outside of a GC? Or are they suggesting that a memory B cell leaving one GC may re-enter a nearby GC without leaving the LN? It is unclear how the authors would distinguish GCs seeded by a memory B cell that has not left the LN from a GC seeded by recirculating memory B cells. You suggest that "PA-GFP activation and sequencing could provide greater clarity", but it is unclear how this could distinguish between these two possibilities. Sequencing provides definitive proof of clonal relationships between cells but is inherently incompatible with further longitudinal imaging. PA-GFP has a short half-life in rapidly dividing cells, and to demonstrate that one labeled cell productively seeds a GC, it must not simply be observed inside a given GC but must be observed to divide in this GC. You should clarify which mechanism(s) you believe to be responsible for this phenomenon and why. While it may be beyond the scope of the current paper to fully solve this issue, is there a way forward statistically or through direct observation to resolve what is happening using this method?

2) You note that some GCs exhibit a clonal "devolution", in which a relatively monoclonal GC becomes less monoclonal. "Devolution" or "stagnation" are problematic, misleading ways to think about this process, as they do not consider the function of the GC: to generate high affinity antibodies. The finding that GCs do not monotonically progress towards monoclonality is important but reflects in this case simply an early clone being outcompeted by a clone with a higher affinity for antigen, not a regressive process.

It is also possible that over the course of a chronic immune reaction, additional antigens may be presented on FDCs. B cells recognizing these new antigens would also compete for T cell help with B cells recognizing the "original" antigens. Changes in the Tfh populations inside the GC over the course of a chronic immune reaction would be unsurprising. All of these would lead to replacement of one population of cells by another, independent of the prevalence of the major clone inside the GC.

Taken together, without measuring and documenting a decrease in antibody affinity (ideally from individual cells), it is misleading to talk about "devolution" or "stagnation".

3) You should show that the afferent and efferent lymphatic drainage is unaffected by the implantation of the chronic imaging window. This could be done by injection of fluorescent dextrans and imaging their appearance in the implanted LN, and in the lymphatic vasculature.

4) As the imaging window implantation causes inflammation, absolute cell counts per LN are a more appropriate comparison than percentages. The authors should provide absolute cell counts too. Further, the gross morphological assessment (i.e. size, weight) should be presented quantitatively, not anecdotally. Additional information about lymphocyte phenotype would also be informative (e.g. expression of CD44, CD62L, and CCR7).

5) You examined granulocytes and lymphocytes, but not monocyte/macrophages and dendritic cells in implanted versus control and sham-implanted LNs. These cells would be expected to be recruited by inflammation – i.e. the implanted LN – and could affect the subsequent kinetics of any immune reaction. You should demonstrate that these cells are not recruited and should present both percentages and absolute cell counts.

6) The authors acknowledge subtle changes in lymphocyte populations in implanted, as compared to sham-operated LNs (decrease in CD4+, CD8+ T cells, increase in B220+ cells, increased GC B cells). However, they do not adequately explain what could be responsible for these changes, and how these processes could also affect the germinal center reaction.

7) The data presented in Figure 4 to show the fidelity of clonal development in the CLNW mice are somewhat unclear. First, two groups of two mice are small, and would require a dramatic difference to show statistical nonequivalence p < 0.05. The authors should calculate what the power of this number of animals is, i.e. what difference they are capable of detecting.

8) It is further not clear to me that the BM chimera ARTEMIS model and the NP-CGG model are similar enough that they can be grouped together as a total n=4, instead of considered independently in separate panels, each with n=2. They should certainly be color-coded separately in Figure 4B-D. This problem is compounded by plotting one number per mouse. The authors should plot individual GCs, denoting which GCs belong to which mouse, in order to increase the number of plotted data points. This would demonstrate the variance between individual GCs in the same animal, which is valuable and informative information. Also, as the divergence is calculated along both X and Y axes (for window LN and explant LN), shouldn't there also be error bars along the X axis for the divergence in the window LN?

---

## [Author Response]

Essential revisions:1) One of the most important applications of the method is the idea that B cell clones may move between GC, which is thought not to take place. You note that in some cases neighbouring GCs exhibit similar color changes at the same time and suggest that this may be due to GC trafficking. The proposed mechanism is unclear – do the authors have evidence that GC cells are found outside of a GC? Or are they suggesting that a memory B cell leaving one GC may re-enter a nearby GC without leaving the LN? It is unclear how the authors would distinguish GCs seeded by a memory B cell that has not left the LN from a GC seeded by recirculating memory B cells. You suggest that "PA-GFP activation and sequencing could provide greater clarity", but it is unclear how this could distinguish between these two possibilities. Sequencing provides definitive proof of clonal relationships between cells but is inherently incompatible with further longitudinal imaging. PA-GFP has a short half-life in rapidly dividing cells, and to demonstrate that one labeled cell productively seeds a GC, it must not simply be observed inside a given GC but must be observed to divide in this GC. You should clarify which mechanism(s) you believe to be responsible for this phenomenon and why. While it may be beyond the scope of the current paper to fully solve this issue, is there a way forward statistically or through direct observation to resolve what is happening using this method?

We thank the reviewer for this very elegant consideration, which we admittedly had not discussed in the manuscript. In response to this point, we have added a discussion of the potential of the described method to resolve these important questions. Recent work by Gabriel Victora’s group, and earlier work by numerous other groups, has indeed indicated that GC B cells are unlikely to move between GC. Therefore, we have focused the discussion on the likelihood of auto-reactive B memory cells returning to GC for further maturation, as previously reported by McHeyzer-Williams for continued diversification of B memory cells specific for foreign antigen. In this context, it is relevant to note that the autoreactive GC ‘never rests’, as self-antigen is ubiquitous and can never be fully cleared. In line with this notion, in our mixed chimeric model (Degn et al., 2017) we have not seen resting long-term memory cells in the bone marrow (Degn and Carroll, unpublished results). As a general approach to address the question of cyclic B memory cell exit and reentry in GCs, we propose employing anti-CD40L blockade of GCs in tamoxifen-pulsed aid-YFP 564 mice and studying reemergence of YFP+ GC B cells after clearance of the anti-CD40L antibody and reformation of GCs. An alternative approach could be through adoptive transfer of sorted aid-YFP memory cells to 564 mice. However, these approaches would not answer whether such reentry occurs without the memory B cell leaving the node first, as would be suggested by our specific observation of ‘color synchrony’ between neighboring GCs. To address this particular question, we would suggest employing a strategy similar to that used by Gabriel Victora and Michel Nussenzweig’s groups to elucidate T follicular helper cell dynamics in GCs (Shulman et al., 2013), combined with our new window approach. Shulman et al. successfully demonstrated that Tfh can traffic between GCs, using conventional intravital lymph node imaging and photoactivation, followed by closure of the incision and follow-up by explant analyses of the same nodes. Immediately after photoactivation, PA-GFP+ Tfh cells were restricted to the targeted region, but ~20 hours after activation, around one third of photoactivated Tfh cells were found outside the original GC. This trafficking only occurred to a significant extent between GCs within the same node on the relatively short time scales analyzed. Only a very small number of photoactivated cells could be detected in the blood or pooled distal lymphoid organs of mice examined 36 hours after photoactivation. One advantage in that study was that Tfh cells are in G1 phase and hence do not dilute out PA-GFP by division. Of note, the half-life of photoactivated PA-GFP in naive B cells was estimated to be 30 hours (Victora et al., 2010), indicating that the protein itself is indeed stable enough on these time-scales. A disadvantage to the application presented here is of course that dark zone GC B cells are rapidly proliferating. However, these cells would not be expected to directly give rise to short-term memory B cells (or other output cells), as mutated GC B cells have to undergo an affinity selection step in the light zone. Thus, we would expect the (progenitor) memory B cells to exit non-concomitantly with division, minimizing concerns of dilution of PA-GFP. Of course, should these memory B cells be reactivated and participate in the GC, they would begin dividing and diluting out the PA-GFP. These processes can be followed longitudinally locally with the new window setup and combining this with explants of distal lymphoid tissue would address whether significant numbers of memory B cells exit GCs.

2) You note that some GCs exhibit a clonal "devolution", in which a relatively monoclonal GC becomes less monoclonal. "Devolution" or "stagnation" are problematic, misleading ways to think about this process, as they do not consider the function of the GC: to generate high affinity antibodies. The finding that GCs do not monotonically progress towards monoclonality is important but reflects in this case simply an early clone being outcompeted by a clone with a higher affinity for antigen, not a regressive process.It is also possible that over the course of a chronic immune reaction, additional antigens may be presented on FDCs. B cells recognizing these new antigens would also compete for T cell help with B cells recognizing the "original" antigens. Changes in the Tfh populations inside the GC over the course of a chronic immune reaction would be unsurprising. All of these would lead to replacement of one population of cells by another, independent of the prevalence of the major clone inside the GC.Taken together, without measuring and documenting a decrease in antibody affinity (ideally from individual cells), it is misleading to talk about "devolution" or "stagnation".

We really appreciate the reviewer’s thoughts regarding devolution, stagnation, and the nature of GC evolution. We agree that the complex interplay between antigen libraries and presentation, somatic hypermutation and possible reseeding, as well as the availability of Tfh, all contribute to the dynamic GC population and peculiar kinetics we have observed. Further, we think the continuous observation of clonal dynamics within a given GC as reflective of the sum of intrinsic B cell and GC environmental processes is very interesting and an improvement in terms of thinking of what is being measured in changes in clonality. The intent of “devolution” in the manuscript was not to imply a decreased affinity per se but rather to describe the ‘regression’ from mono-/oligo-clonality (as inferred by color, which is admittedly a rough approximation) towards increasing ‘color diversity’. As such we agree with the reviewer and a more appropriate term may be ‘inversion’. We have edited the manuscript accordingly, omitting the term "devolution", and have included a discussion of the above point. In this connection, we have also discussed recent elegant work by Garnett Kelsoe’s group on responses to complex antigen and put this into the perspective of Michel Nussenzweig’s papers on the ‘open nature’ of GCs, where he demonstrated that mainly B cells responding to the same antigen or catering to the same T helper cells, would be able to enter an ongoing GC. This suggests that a GC elicited in response to antigen A is rarely taken over by clones responding to antigen B, if A and B are unrelated. Of course, the chronic scenario may differ from the transient scenario, as also indicated by the reviewer.

3) You should show that the afferent and efferent lymphatic drainage is unaffected by the implantation of the chronic imaging window. This could be done by injection of fluorescent dextrans and imaging their appearance in the implanted LN, and in the lymphatic vasculature.

This is also an excellent point. In the preparation of the B1-8hi early GC experiment we actually conducted several control experiments. We injected rabbit-anti-PE antibody 24 hours prior to the surgery, and then injected PE either just prior to surgical implantation or following surgical implantation of the chamber. In either case, intact afferent lymphatics are required for robust labeling of FDCs in the inguinal nodes. We noted robust and similar staining of FDC networks regardless of whether we injected prior to, or just after surgical implantation of the window chamber. There was only a minimal and statistically non-significant difference between pre- and post-implant injection of PE 6 hours after surgery, and no difference 24 hours after surgery. Along with the fact that technically (surgical approach) there is minimal perturbation of the deep and peripheral afferent lymphatics these data support intact afferent drainage to operated nodes. Lack of peripheral edema in any of dozens of surgeries is strong support of lack of injury to the efferent lymphatic drainage. We have included these additional data as Figure 2—figure supplement 1 and a discussion of this important point in the manuscript.

4) As the imaging window implantation causes inflammation, absolute cell counts per LN are a more appropriate comparison than percentages. The authors should provide absolute cell counts too. Further, the gross morphological assessment (i.e. size, weight) should be presented quantitatively, not anecdotally. Additional information about lymphocyte phenotype would also be informative (e.g. expression of CD44, CD62L, and CCR7).

This is another good point. Ideally, we should have included counting beads for absolute quantification, but this was not done. However, we have the volumes of FACS samples and the total numbers of spleen cells available, which has allowed us to give a close estimate of absolute counts instead of relative percentages. We unfortunately do not have weights or height/widths of the nodes, as we noted minimal perturbation to LN morphology throughout these experiments, and hence did not consider it for a more comprehensive analysis. Please also refer to our considerations regarding point 5, below.

5) You examined granulocytes and lymphocytes, but not monocyte/macrophages and dendritic cells in implanted versus control and sham-implanted LNs. These cells would be expected to be recruited by inflammation – i.e. the implanted LN – and could affect the subsequent kinetics of any immune reaction. You should demonstrate that these cells are not recruited and should present both percentages and absolute cell counts.

We appreciate this point, and again we must concede that we have not done all potentially interesting analyses. However, in the analyses we did perform, we actually used Gr1, which recognizes both Ly6C (mainly monocytes) and Ly6G (neutrophils), and therefore should provide a reasonable estimate of inflammatory cell recruitment. Indeed, as is evident from the relative percentages and now cell counts presented in the present version of the manuscript, there is a slight influx of these cells into the window node, and a small upregulation in the spleen. As for above, we have of course provided cell counts instead of relative frequencies to give a more accurate depiction of this effect in agreement with the reviewer’s comment. We cannot argue that there is not a low-level degree of inflammation in the lymph node associated with the window implantate, either as an effect of sterile inflammation due to the opening of the overlying tissue or due to the increased exposure to skin-associated commensals at the incision site, or both. Unfortunately, it is an inherent problem to any procedure breaking the skin barrier, but we have sought to minimize it as much as possible and based on our results we do not think that it impacts our ability to assess antigen-specific clonal responses in germinal centers. In the updated manuscript, we have included the cell counts for the Gr1+ cells and explain the subsets represented by this marker and present the above considerations.

6) The authors acknowledge subtle changes in lymphocyte populations in implanted, as compared to sham-operated LNs (decrease in CD4+, CD8+ T cells, increase in B220+ cells, increased GC B cells). However, they do not adequately explain what could be responsible for these changes, and how these processes could also affect the germinal center reaction.

Granted, there is some low-level inflammation, as the window cannot with certainty be 100% sterile and there may be some possibility of increased antigen diversity. We have now included a discussion of the relevant distinction between ‘sterile inflammation’ (tissue damage and exposure) versus non-sterile inflammation (skin commensals and whatever else could potentially access the chamber), where the former might not be as critical as the latter. However, based on our studies of clonality in the NP-CGG and 564 models, this does not dramatically impact evolution of the GCs. Please also refer to our response to point 7, below.

7) The data presented in Figure 4 to show the fidelity of clonal development in the CLNW mice are somewhat unclear. First, two groups of two mice are small, and would require a dramatic difference to show statistical nonequivalence p < 0.05. The authors should calculate what the power of this number of animals is, i.e. what difference they are capable of detecting.

We have added a new panel (Figure 4B), displaying raw data for GC color dominance on a per-GC, per-mouse basis. We also added x-dimension error bars to Figure 4C-E. Regarding the question of statistical non-equivalence, this is rather based on the total number of observed GC than the number of mice. Increasing the number of mice would perhaps allow us to determine the frequency of mice in which color dominance would deviate between window implant and contralateral side, but this would merely be an expression of the frequency of failure of the surgery/chamber implant (causing perturbation of homeostasis in the implanted node), and as such refers more to a technical parameter (this would vary depending on the experience and proficiency of the surgeon, but would not necessarily reflect on the adequacy of the method itself). Having a relatively high number of observed GCs within each mouse allows a statistically sound interpretation of whether implanted and contralateral node GCs deviate. Increasing this number would only be predicted to decrease noise/variation in individual GC color dominance (as estimated based on the large set of explants analyzed in Degn et al., 2017). Therefore, we strongly believe that our statistical analysis is sound. Please also see the response to point 8, below, regarding pooling ARTEMIS and NP-CGG mice. Furthermore, based on the following considerations, we do not believe that a power analysis will change this conclusion.

We appreciate the reviewer’s concern in distinguishing between true negatives and false negatives. It is an important consideration alongside any statistical analysis. Unfortunately, a post-hoc power analysis would not provide additional power in the presented analysis, since p-values are directly related to the observed power (please refer Figure 1, Hoenig and Heisey, 2001). From this, we can see that there is always low observed power associated with non-significant (p>0.05) tests. We believe that the emphasis on conclusive interpretation of statistical tests disregards the hypothesis-based nature of these tests. That is, a p-value of 0.1 is not “stronger” than 0.3 despite the fact that the observed power would necessarily be greater, and this is independent of sample size. The cut-off of 0.05 is merely a convention, and we would argue for placing relatively greater emphasis on our confidence intervals for effect sizes.

In the case of our Figure 5C-E (most relevant to the current discussion), we believe that even if the “true” β coefficients were at the extremes of the confidence intervals, this would not be great cause for concern, for the following reasons. For both Figure 5C and D it seems unlikely that a deviation within 4-8% of true color domination represents a meaningful departure. That is to say, in the event we increased the sample size, it is unlikely that these intervals would depart from unity towards the extremes. In the case of Figure 5E, the interval could be more concerning. A true deviation of ~30-40% from the unperturbed scenario would significantly reduce our confidence in this technique. However, it is important to note the large variation in the divergence index (as compared to the other measures of GC dynamics). The divergence index can be significantly skewed as a result of rarer color combinations (such as YFP/CFP or GFP) becoming dominant clones in GC (Degn et al., 2017, and Tas et al., 2016). In this case it is more likely that the width of the confidence interval reflects the underlying variation associated with the divergence index rather than a true observation of failure of this technique to recapitulate homeostatic conditions. This notion is supported by Figure 5C and D, and the other lines of evidence presented, which suggest that homeostasis is not significantly perturbed.

8) It is further not clear to me that the BM chimera ARTEMIS model and the NP-CGG model are similar enough that they can be grouped together as a total n=4, instead of considered independently in separate panels, each with n=2. They should certainly be color-coded separately in Figure 4B-D. This problem is compounded by plotting one number per mouse. The authors should plot individual GCs, denoting which GCs belong to which mouse, in order to increase the number of plotted data points. This would demonstrate the variance between individual GCs in the same animal, which is valuable and informative information. Also, as the divergence is calculated along both X and Y axes (for window LN and explant LN), shouldn't there also be error bars along the X axis for the divergence in the window LN?

As outlined above, we have added a new panel (Figure 4B), displaying raw data for GC color dominance on a per-GC, per-mouse basis. We also added x-dimension error bars to Figure 4C-E. Regarding the question of whether the ARTEMIS and NP-CGG models are similar enough to pool them for the analysis, we note that although these models are very different, one reflecting autoreactive responses to complex antigen, the other a response to a less complex foreign antigen, they were demonstrated to have overall similar clonal evolution kinetics by traditional time-point analyses using the Confetti model (Degn et al., 2017). As suggested by the reviewer, for clarity we have indicated in Figure 4C-E which group the mice were derived from.